# CirT: Global Subseasonal-to-Seasonal Forecasting with Geometry-inspired Transformer

**Yang Liu**[1,2]* **Zinan Zheng**[1]* **Jiashun Cheng**[1,2], **Fugee Tsung**[1,2], **Deli Zhao**[3], **Yu Rong**[3]†, **Jia Li**[1,2]†

[1]The Hong Kong University of Science and Technology (Guangzhou)
[2]The Hong Kong University of Science and Technology  [3]DAMO Academy, Alibaba Group

## Abstract

Accurate Subseasonal-to-Seasonal (S2S) climate forecasting is pivotal for decision-making including agriculture planning and disaster preparedness but is known to be challenging due to its chaotic nature. Although recent data-driven models have shown promising results, their performance is limited by inadequate consideration of geometric inductive biases. Usually, they treat the spherical weather data as planar images, resulting in an inaccurate representation of locations and spatial relations. In this work, we propose the geometric-inspired Circular Transformer (CirT) to model the cyclic characteristic of the graticule, consisting of two key designs: (1) Decomposing the weather data by latitude into circular patches that serve as input tokens to the Transformer; (2) Leveraging Fourier transform in self-attention to capture the global information and model the spatial periodicity. Extensive experiments on the Earth Reanalysis 5 (ERA5) reanalysis dataset demonstrate our model yields a significant improvement over the advanced data-driven models, including PanguWeather and GraphCast, as well as skillful ECMWF systems. Additionally, we empirically show the effectiveness of our model designs and high-quality prediction over spatial and temporal dimensions. The code link is: https://github.com/compasszzn/CirT.

## 1 Introduction

Subseasonal-to-seasonal (S2S) forecasting, which predicts meteorological variables 2 to 6 weeks in advance, is crucial for agriculture, resource allocation, and disaster preparedness (e.g., heatwaves and droughts) (Mouatadid et al., 2024). Despite its high socioeconomic benefits, such a task has long been considered a "predictability desert" (Vitart et al., 2012) due to the chaotic nature of the atmosphere. Compared with medium-range (up to 15 days) and seasonal predictions (3-6 months) (Vitart et al., 2017), the S2S timescale is long enough to lose much of the memory of atmospheric initial conditions, while it is too short for slowly evolving earth system components such as the ocean that strongly influence the atmosphere (Black et al., 2017; Phakula et al., 2024). The existing S2S near-real-time forecasting models rely on physics-based Numerical Weather Prediction (NWP) models that discretize governing equations of thermodynamics, fluid flow, etc (Nathaniel et al., 2024). However, these models generally suffer considerable biases (Mouatadid et al., 2023) and require massive computational resources to perform numerical integration at fine-grained resolutions (Schneider et al., 2023).

Multiple studies utilize the potential of data-driven models to mitigate the above weakness, in which most works (Hwang et al., 2019; He et al., 2022; Mouatadid et al., 2024) focus on regional forecasting. However, the regional weather is often influenced by conditions in other areas on the S2S timescale, indicating the insufficiency of relying solely on regional inputs for S2S forecasting (Vitart et al., 2012; Lau & Waliser, 2011; Robertson et al., 2015). With the development of the high-quality Earth Reanalysis 5 (ERA5) dataset (Hersbach et al., 2020) and weather foundation models (Pathak et al., 2022; Lam et al., 2022; Bi et al., 2023), a few studies (Chen et al., 2024; Nguyen et al., 2023; Weyn et al., 2021) have proposed global data-driven S2S forecasting models and achieved promising results. Specifically, they treat weather parameter values on the latitude-longitude grid (i.e.,

---

*Equal contribution.
†Corresponding authors (jialee@ust.hk, yu.rong@hotmail.com)

graticule) as image data, represented as 3-dimensional tensors, and employ the Transformer (Dosovitskiy, 2020) to forecast the future weather parameter values as an image generation task. Despite the promising results, the inconsistency between the planar and sphere geometry leads to significant distortions in learning dynamics, resulting in incorrect spatial relations. Figure 1 depicts the example of this heavy distortion in planar projection.

Therefore, we re-investigate the transformer design for graticule by considering two geometric inductive biases. First, existing methods decompose the planar latitude-longitude image into the fixed-degree patch, such as $3° \times 3°$, ignoring that parallels have unequal geometric lengths. For example in Figure 1, although the patch size is the same in the planar view, the area of Patch 2 is significantly larger than that of Patch 1. Thus, the generated patches are of varying sizes and shapes in the sphere, especially in high-altitude regions, leading to an uneven distribution of information across patches. Second, the graticule demonstrates latitudinal spatial periodicity. Overlooking such an inductive bias results in inaccurate spatial relation modeling. As shown in Figure 1, the left and right boundaries of the planar view are connected while appear separated.

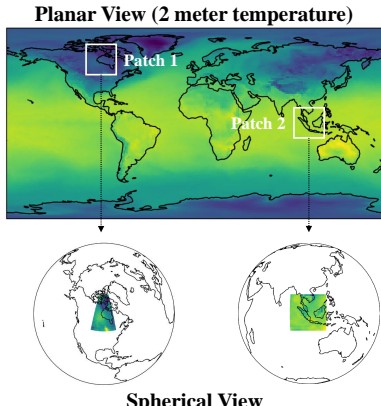

**Planar View (2 meter temperature)**

**Spherical View**

Figure 1: Planar and the spherical view of 2-metre temperature. Treating it as an image results in distortion.

In this work, we propose a Circular Transformer (CirT) that implements an equidistant circular patching strategy and self-attention incorporating spatial periodicity. To construct undistorted spatial relations among patches, CirT partitions the graticule uniformly by latitude, treating weather variables distributed on each parallel as a patch. Thus, the generated patches are in the same shape with geometric lengths determined by latitudes. Meanwhile, the adjacent patches are equidistant. Considering the weather signals are spatially periodic on the circular patch, we treat it as a spatial signal of $2\pi$ periodicity and leverage the Fourier transform to extract the global features and perform self-attention to mix patches on the frequency domain. The frequencies are inversely transformed into the spatial domain to make the final forecasting. Finally, instead of learning autoregressive models like previous works (Chen et al., 2024; Nguyen et al., 2023; Weyn et al., 2021), we directly train CirT to predict in S2S timescale, which avoids the large accumulation errors and learns the connections between the initial and target states. Through extensive experiments on the ERA5 dataset, we find that

- Remarkably, CirT outperforms skillful numerical S2S systems including UKMO, NCEP, CMA, and ECMWF, as well as state-of-the-art data-driven models including ClimaX, FourCastNetV2, PanguWeather, and GraphCast on S2S forecasting.

- Most methods, including data-driven and numerical models, achieve a larger bias in high-latitude areas. In contrast, we show that CirT produces more structurally consistent results with ground truth and performs better in these areas.

- Ablation studies show that the proposed two simple designs, circular patching and patch mixing in the frequency domain, significantly enhance the model performance.

## 2 PRELIMINARY

**Problem Definition** We study the bi-weekly forecasting of $K$ weather parameters at the latitude-longitude grid $\mathcal{G} \in \mathbb{R}^{H \times W \times 2}$. $H$ and $W$ are the height and width of the grid that depends on the resolution of latitude and longitude, and $\mathcal{G}_{h,w,:} = (\lambda_h, \phi_w) \in \Omega = [-90°, 90°] \times [-180°, 180°]$. At day $t$, the state of global weather is represented by a 3-dimensional tensor $\mathcal{X}_t \in \mathbb{R}^{H \times W \times K}$. Following previous works (Chen et al., 2024; Mouatadid et al., 2023; Nguyen et al., 2023), given the initial condition $(\mathcal{G}, \mathcal{X}_{t_1})$, our objective is to learn a neural network to predict the average value of weather variables over weeks 3-4 and weeks 5-6, as shown in the following:

$$(\hat{\mathcal{X}}_{t_{15}:t_{28}}, \hat{\mathcal{X}}_{t_{29}:t_{42}}) = f_\Theta(\mathcal{G}, \mathcal{X}_{t_1}), \tag{1}$$

where $\Theta$ denotes the parameters of neural networks. $\hat{\mathcal{X}}_{t_{15}:t_{28}}$ and $\hat{\mathcal{X}}_{t_{29}:t_{42}}$ are predicted average value over weeks 3-4 (from day 15 to day 28) and weeks 5-6 (from day 29 to day 42). Distinct from

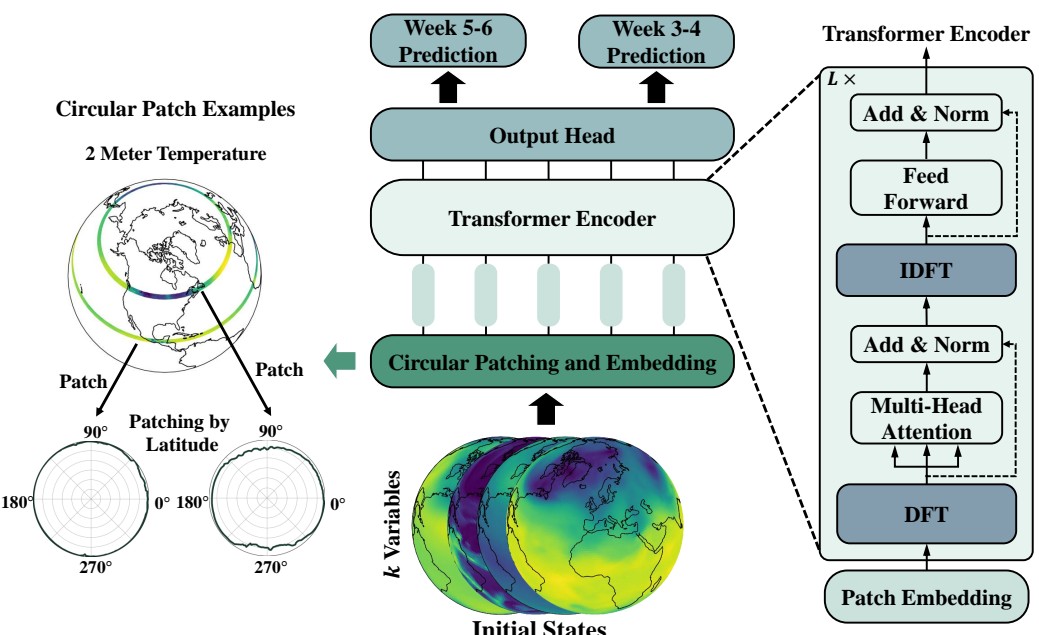

Figure 2: CirT architecture and circular patching examples. The input tensors are first decomposed by latitudes, resulting in a set of circular patches. Then they are fed into a series of Transformer blocks where DFT and IDFT are applied in each block to transform information between frequency and spatial domain. Finally, the output head maps the representation to biweekly predictions.

data-driven medium-range models that iteratively produce the results, we aim to learn a model that directly predicts these two values.

**Fourier Transform and Inversion**  Fourier Transform is known to be an effective tool to extract features from periodic signals. Consider a sequence of $N$ grid-based real-valued observations of a function, denoted by $\boldsymbol{s} = (s_1, ..., s_N) \in \mathbb{R}^N$. The Discrete Fourier Transform (DFT), represented by $\mathcal{F}$, converts this sequence into the frequency domain with a periodicity of $2\pi$ as follows:

$$S_k = \sum_{n=1}^{N} s_n \cos\left(2\pi \frac{k}{N} n\right) - i \sum_{n=1}^{N} s_n \sin\left(2\pi \frac{k}{N} n\right) = A_k - B_k i, \tag{2}$$

where $\boldsymbol{S} = \mathcal{F}(\boldsymbol{s}) = (S_1, ..., S_N) \in \mathbb{R}^N$ and $i$ is the imaginary unit. $A_k$ and $B_k$ are the real and imaginary parts of the complex number $S_k$ in the frequency domain, respectively. The inverse transformation, which reconstructs the original sequence from the frequency domain, is given by:

$$s_n = \frac{1}{N} \sum_{k=1}^{N} S_k \left(\cos\left(2\pi \frac{n}{N} k\right) + i \sin\left(2\pi \frac{n}{N} k\right)\right), \tag{3}$$

or equivalently by substituting $S_k = A_k - B_k i$ ,

$$s_n = \frac{1}{N} \sum_{k=1}^{N} \left(A_k \cos\left(2\pi \frac{n}{N} k\right) - B_k \sin\left(2\pi \frac{n}{N} k\right)\right), \tag{4}$$

where the imaginary unit is canceled out. We express the inverse transform as $\boldsymbol{s} = \mathcal{F}^{-1}(\boldsymbol{S})$ for symmetry.

## 3  CIRT MODEL

Our CirT architecture is illustrated in Figure 2. The input $\boldsymbol{\mathcal{X}}_{t_1}$ is split to $H$ embedded circular patches and then fed into the Transformer encoder to predict the Weeks 3-4 and Weeks 5-6 results. In the following, we elaborate on the model structure including circular patching and transformer encoder.

**Circular Patching** CirT first divides the input $\mathcal{X}_{t_1}$ into $H$ latitudinal non-overlapping patches $\{\boldsymbol{X}^{(h)}\}_{h=1}^{H}$ where $\boldsymbol{X}^{(h)} \in \mathbb{R}^{W \times K}$ and its $w$-th row $\boldsymbol{X}_w^{(h)} \in \mathbb{R}^K$ denotes the $K$ weather values at the coordinate $(\lambda_h, \phi_w)$. Thus, the geometric distance of adjacent patches $\boldsymbol{X}_w^{(h)}$ and $\boldsymbol{X}_{w+1}^{(h)}$ is fixed to $R\Delta\phi$ where $\Delta\phi$ is the longitude resolution and $R$ is the earth radius. Meanwhile, the geometric length of the patch $\boldsymbol{X}^{(h)}$ can be determined by $2\pi R \cos(\lambda_h)$. These patches are then flattened and stacked into a matrix $\boldsymbol{X}^F \in \mathbb{R}^{H \times (W \cdot K)}$, which is subsequently projected into latent space to generate initial embedding $\boldsymbol{E} \in \mathbb{R}^{H \times D}$ as follows:

$$\boldsymbol{E} = \boldsymbol{X}^F \boldsymbol{W}_p + \boldsymbol{W}_{pos}, \tag{5}$$

where $\boldsymbol{W}_p \in \mathbb{R}^{(W \cdot K) \times D}$ and $\boldsymbol{W}_{pos} \in \mathbb{R}^{H \times D}$ denote the learnable projection matrix and additive position embedding, respectively. Subsequently, the initial embedding is fed into the Transformer encoder for further processing.

**CirT Transformer Encoder** Considering the circular patch satisfies $X_w = X_{w+W}$. Therefore, instead of directly inputting $X$ into the transformer, we consider its Fourier transform, composed of coefficients from a series of periodic basis functions. The Fourier Transform offers insights into the wave frequencies present in the periodic signals, which aligns well with data with inherent periodicities (Zhou et al., 2022; Wu et al., 2023). Our approach aims to incorporate such an inductive bias into the learning process by operating in the frequency domain. Specifically, at the $l$-th transformer block, we begin by applying DFT to each row of input embedding $\boldsymbol{E}^{(l)} \in \mathbb{R}^{H \times D}$, where $\boldsymbol{E}_h^{(l)} = (E_{h,1}^{(l)}, ..., E_{h,D}^{(l)}) \in \mathbb{R}^D$ corresponds to the embedding of the $h$-th patch. According to Eqn. 2,

$$S_{h,k}^{(l)} = \sum_{n=1}^{D} E_{h,n}^{(l)} \cos\left(2\pi \frac{k}{N} n\right) - i \sum_{n=1}^{D} E_{h,n}^{(l)} \sin\left(2\pi \frac{k}{N} n\right) = A_{h,k}^{(l)} - B_{h,k}^{(l)} i, \tag{6}$$

where $\boldsymbol{S}_h^{(l)} = \mathcal{F}(\boldsymbol{E}_h^{(l)}) \in \mathbb{R}^D$ represents the complex frequency embedding, and $\boldsymbol{A}_h^{(l)} = \mathrm{Re}(\boldsymbol{S}_h^{(l)})$ and $\boldsymbol{B}_h^{(l)} = \mathrm{Im}(\boldsymbol{S}_h^{(l)})$ represent its real and imaginary parts, respectively. These components of all patches are then stacked into matrices $\boldsymbol{A}^{(l)} \in \mathbb{R}^{N \times D}$ and $\boldsymbol{B}^{(l)} \in \mathbb{R}^{N \times D}$, which are then jointly fed to the multi-head attention to consider their correlation.

For the $m$-th attention head, we compute the query, key, and value matrices of $\boldsymbol{C}^{(l)} = [\boldsymbol{A}^{(l)}, \boldsymbol{B}^{(l)}] \in \mathbb{R}^{H \times 2D}$ following standard attention operations:

$$\boldsymbol{Q}^{(l,m)} = \boldsymbol{C}^{(l)} \boldsymbol{W}_m^Q, \quad \boldsymbol{K}^{(l,m)} = \boldsymbol{C}^{(l)} \boldsymbol{W}_m^K, \quad \boldsymbol{V}^{(l,m)} = \boldsymbol{C}^{(l)} \boldsymbol{W}_m^V, \tag{7}$$

where $\boldsymbol{W}_m^{Q/K/V} \in \mathbb{R}^{2D \times 2D}$ are learned projection matrices. The attention output $\tilde{\boldsymbol{A}}^{(l,m)}$ and $\tilde{\boldsymbol{B}}^{(l,m)}$ are then computed using scaled production:

$$\tilde{\boldsymbol{C}}^{(l,m)} = [\tilde{\boldsymbol{A}}^{(l,m)}, \tilde{\boldsymbol{B}}^{(l,m)}] = \mathrm{softmax}\left(\frac{\boldsymbol{Q}^{(l,m)} \boldsymbol{K}^{(l,m)}}{\sqrt{D}}\right) \boldsymbol{V}^{(l,m)}, \tag{8}$$

where $\tilde{\boldsymbol{A}}^{(l,m)}$ consists of the first $D$ columns of the output $\tilde{\boldsymbol{C}}^{(l,m)}$ while $\tilde{\boldsymbol{B}}^{(l,m)}$ corresponds to the rest. For the $h$-th patch, the processed frequency-domain representation is then rearranged as $\tilde{\boldsymbol{S}}_h^{(l,m)} = \tilde{\boldsymbol{A}}_h^{(l,m)} - \tilde{\boldsymbol{B}}_h^{(l,m)} i$. This representation is then converted back to the original domain by applying the inverse DFT following Eqn. 4:

$$\tilde{E}_{h,n}^{(l,m)} = \frac{1}{D} \sum_{k=1}^{D} \left( \tilde{A}_{h,k}^{(l,m)} \cos\left(2\pi \frac{n}{N} k\right) - \tilde{B}_{h,k}^{(l,m)} \sin\left(2\pi \frac{n}{N} k\right) \right), \tag{9}$$

where $\tilde{\boldsymbol{E}}_h^{(l,m)} = \mathcal{F}^{-1}(\tilde{\boldsymbol{S}}_h^{(l,m)})$ is the inverse-transformed representation of the patch. By stacking these transformed representations, we obtain the reconstructed embedding $\tilde{\boldsymbol{E}}^{(l,m)}$. Finally, a feed-forward network is used to generate the output embedding $\boldsymbol{E}^{(l+1)}$ by passing the reconstructed embeddings from all heads:

$$\boldsymbol{E}^{(l+1)} = \mathrm{MLP}([\tilde{\boldsymbol{E}}^{(l,1)}, \cdots, \tilde{\boldsymbol{E}}^{(l,M)}]). \tag{10}$$

For brevity, we omit the LayerNorm layers in the forward process of the Transformer block, as shown in Figure 2.

**Training** Given the output representation of the Transformer encoder, a flattened MLP is used to obtain the prediction results of weeks 3-4 and weeks 5-6. The model is trained to minimize the discrepancy between the prediction and ground truth. The loss in each variable and location is gathered and averaged over weeks 3-4 and 5-6 to calculate the overall objective loss:

$$\mathcal{L} = \frac{1}{2 \times K \times H \times W} (||\hat{\boldsymbol{\mathcal{X}}}_{t_{15}:t_{28}} - \boldsymbol{\mathcal{X}}_{t_{15}:t_{28}}||_2^2 + ||\hat{\boldsymbol{\mathcal{X}}}_{t_{29}:t_{42}} - \boldsymbol{\mathcal{X}}_{t_{29}:t_{42}}||_2^2), \tag{11}$$

where $\boldsymbol{\mathcal{X}}_{t_{15}:t_{28}}$ and $\boldsymbol{\mathcal{X}}_{t_{29}:t_{42}}$ are the ground truth.

## 4 EXPERIMENTS

**Dataset** We evaluate the effectiveness of CirT on the ERA5 reanalysis dataset (Hersbach et al., 2020) which provides the comprehensive pressure and single levels climate variables. The resolution is set to $1.5°$, resulting in a $121 \times 240$ latitude-longitude grid. We use 6 pressure level variables, including geopotential ($z$), specific humidity ($q$), temperature ($t$), u component of wind ($u$), v component of wind ($v$), and vertical velocity ($w$) at 10 pressure levels: 10, 50, 100, 200, 300, 500, 700, 850, 925 and 1000 hPa. Besides, we integrate 3 more single levels variables: 2m temperature ($t2m$), 10m u component of wind ($10u$), 10m v component of wind ($10v$), totaling 63 variables. We use the 1979–2016 (38 years of) data for training, the 2017 data for validation, and the 2018 for testing.

**Metric** Following existing works (Rasp et al., 2024; Nathaniel et al., 2024), we adopt latitude-weighted RMSE and Anomaly Correlation Coefficient (ACC) to evaluate the model performance with $K = 1$, which are defined as follows:

$$\text{RMSE} = \sqrt{\frac{1}{HW} \sum_{h,w} \alpha(h)(\hat{\boldsymbol{X}}_{h,w} - \boldsymbol{X}_{h,w})^2}, \ \text{ACC} = \frac{\sum_{h,w} \alpha(h)\hat{\boldsymbol{X}}'_{h,w}\boldsymbol{X}'_{h,w}}{\sqrt{\sum_{h,w} \alpha(h)\hat{\boldsymbol{X}}'^2_{h,w} \sum_{h,w} \alpha(h)\boldsymbol{X}'^2_{h,w}}}, \tag{12}$$

where $\alpha(h) = \cos(\lambda_h)/\frac{1}{H}\sum_{h'}\cos(\lambda_{h'})$ is the latitude weighting factor. $\boldsymbol{X} = \boldsymbol{\mathcal{X}}_{t,:,:,1} \in \mathbb{R}^{H \times W}$ is the ground truth for specific day $t$ with its prediction $\hat{\boldsymbol{X}}$. $\boldsymbol{X}'_{h,w} = \boldsymbol{X}_{h,w} - C$ and $\hat{\boldsymbol{X}}'_{h,w} = \hat{\boldsymbol{X}}_{h,w} - C$, where $C$ is the observational climatology (i.e., empirical mean of observational data).

**Data-driven baselines** Following the existing S2S benchmark (Nathaniel et al., 2024), we compare CirT with state-of-the-art data-driven models, including FourCastNetV2 (Pathak et al., 2022), PanguWeather (Bi et al., 2023), GraphCast (Lam et al., 2022) and ClimaX (Nguyen et al., 2023). Among them, FourCastNetV2, PanguWeather, and ClimaX follow a ViT process, while GraphCast is a graph neural network.

**Physics-based baselines** To further evaluate the model performance of CirT, we compare it with various advanced physics-based models, including UK Meteorological Office (UKMO) (Williams et al., 2015), National Centers for Environmental Prediction (NCEP) (Saha et al., 2014), China Meteorological Administration (CMA) (Wu et al., 2019), European Centre for Medium-Range Weather Forecasts (ECMWF) (Molteni et al., 1996). Among them, ECMWF is recognized as the most skillful S2S modeling system (Chen et al., 2024; Domeisen et al., 2022). More details are shown in the Appendix.

**Implementation details** We use the following hyper-parameters for all direct training baselines: Batch size 16, the hidden dimension 256, and the attention head 16. All models are set to 8 layers and the learning rate is 0.01. All models are trained for 20 epochs. All models are implemented based on Pytorch Lightning, trained on 8 GeForce RTX 4090 GPU. We train ClimaX the same as CirT and download trained FourCastNetV2, PanguWeather, and GraphCast through API[1] provided by ECMWF. We perform the inference of Download models in NVIDIA A800 80G GPU. We use the download parameters and do not finetune them to the S2S timescale due to the limited computational resources. Note that although FourCastNetV2 and PanguWeather report the 2-meter predictions, the retrieved model in ECMWF does not include its inference and GraphCast is out-of-memory when performing inference for Weeks 5-6.

---

[1]https://github.com/ecmwf-lab/ai-models

Table 1: Global S2S forecasting results of data-driven models. The lower RMSE and higher ACC indicate better results.

| Metric | RMSE (↓) | | | | | ACC (↑) | | | | |
|---|---|---|---|---|---|---|---|---|---|---|
| | FourCastNetV2 | GraphCast | PanguWeather | ClimaX | CirT | FourCastNetV2 | GraphCast | PanguWeather | ClimaX | CirT |
| **Weeks 3-4** | | | | | | | | | | |
| z500 (m$^2$/s$^2$) | 615 | 618 | 649 | 602 | **477** | 0.947 | 0.973 | 0.963 | 0.977 | **0.984** |
| z850 (m$^2$/s$^2$) | 402 | 411 | 416 | 372 | **304** | 0.896 | 0.931 | 0.926 | 0.949 | **0.963** |
| t500 ($K$) | 2.093 | 2.176 | 2.271 | 2.186 | **1.687** | 0.966 | 0.979 | 0.966 | 0.981 | **0.988** |
| t850 ($K$) | 2.390 | 2.370 | 2.569 | 2.618 | **1.903** | 0.957 | 0.982 | 0.963 | 0.981 | **0.988** |
| t2m ($K$) | – | 2.158 | – | 2.998 | **2.007** | – | 0.991 | – | 0.985 | **0.993** |
| u10 (m/s) | 2.328 | – | 2.431 | 2.334 | **1.806** | 0.830 | – | 0.812 | 0.817 | **0.896** |
| v10 (m/s) | 1.896 | – | 1.984 | 1.906 | **1.511** | 0.712 | – | 0.686 | 0.667 | **0.811** |
| **Weeks 5-6** | | | | | | | | | | |
| z500 (m$^2$/s$^2$) | 652 | – | 754 | 619 | **471** | 0.943 | – | 0.956 | 0.976 | **0.985** |
| z850 (m$^2$/s$^2$) | 426 | – | 461 | 375 | **301** | 0.889 | – | 0.911 | 0.948 | **0.964** |
| t500 ($K$) | 2.250 | – | 2.829 | 2.254 | **1.672** | 0.963 | – | 0.958 | 0.980 | **0.988** |
| t850 ($K$) | 2.567 | – | 2.998 | 2.741 | **1.933** | 0.953 | – | 0.957 | 0.980 | **0.989** |
| t2m ($K$) | – | – | – | 3.168 | **2.026** | – | – | – | 0.984 | **0.992** |
| u10 (m/s) | 2.479 | – | 2.679 | 2.355 | **1.809** | 0.812 | – | 0.783 | 0.814 | **0.895** |
| v10 (m/s) | 1.980 | – | 2.104 | 1.939 | **1.512** | 0.691 | – | 0.655 | 0.659 | **0.812** |

## 4.1 OVERALL PERFORMANCE

**Compared with data-driven models** We display the model performance in 7 target variables: geopotential at 500hPa ($z500$), geopotential at 850hPa ($z850$), temperature at 500hPa ($t500$), temperature at 850hPa ($t850$), 2m temperature ($t2m$), 10 metre U wind component ($u10$) and 10 metre V wind component ($v10$) in Table 1. Based on these results, we have the following observations:

- CirT consistently outperforms all baselines in all cases. Specifically, compared to the best baseline, the average RMSE improvement on geopotential (m$^2$/s$^2$) and temperature ($K$) over Weeks 3-4 and Weeks 5-6 is $96.5, 0.369$, and $111, 0.843$, demonstrating significant improvement.

- CirT achieves larger improvement over Weeks 5-6 than Weeks 3-4 predictions. The iterative models (i.e., FourCastNetV2 and PanguWeath) accumulate errors in each step, leading to inaccurate predictions when the iterative step is large. In contrast, the direct prediction models aim to capture the relations between initial and subseasonal states, resulting in lower performance drops.

- Compared with ViT-based iterative models, GraphCast achieves relatively better performance in Weeks 3-4 predictions. The reason can be attributed to that it employs mesh to model the sphere geometry, leading to lower accumulated errors.

- We can find that wind forecasting is more challenging than other comparing variables. For example, Weeks 3-4 t850 ACC of FourCastNetV2 is 0.957 while u10 ACC is 0.830. Under such cases, CirT still performs the best, further verifying its effectiveness.

In addition, we provide a relative RMSE comparison in Figure 3. We can observe that CirT generally achieves lower errors across all pressure levels. When the lead time increases from Weeks 3-4 to 5-6, the performance of baselines significantly reduces, especially in Temperature. In contrast, CirT maintains relatively low errors.

**Compared with numerical models** To better investigate the performance of CirT, we compare it and numerical models in Figure 3, where lighter colors indicate lower RMSE. The ACC results are shown in the Appendix. From the results, we can find that:

- CirT remarkably outperforms numerical models in almost all cases, demonstrating the effectiveness of direct data-driven models. In addition, numerical methods underperform at low-pressure levels, especially at the 10 hPa level, while CirT still performs better.

- Compared with the results of Weeks 3-4 and 5-6, we can observe that similar to iterative data-driven models, the performance of numerical models decreases when the lead time increases but the performance drop is smaller than data-driven models. Such results show that physics-based models are more stable than iterative data-driven models. Overall, CirT which directly predicts the biweekly states maintains the best performance.

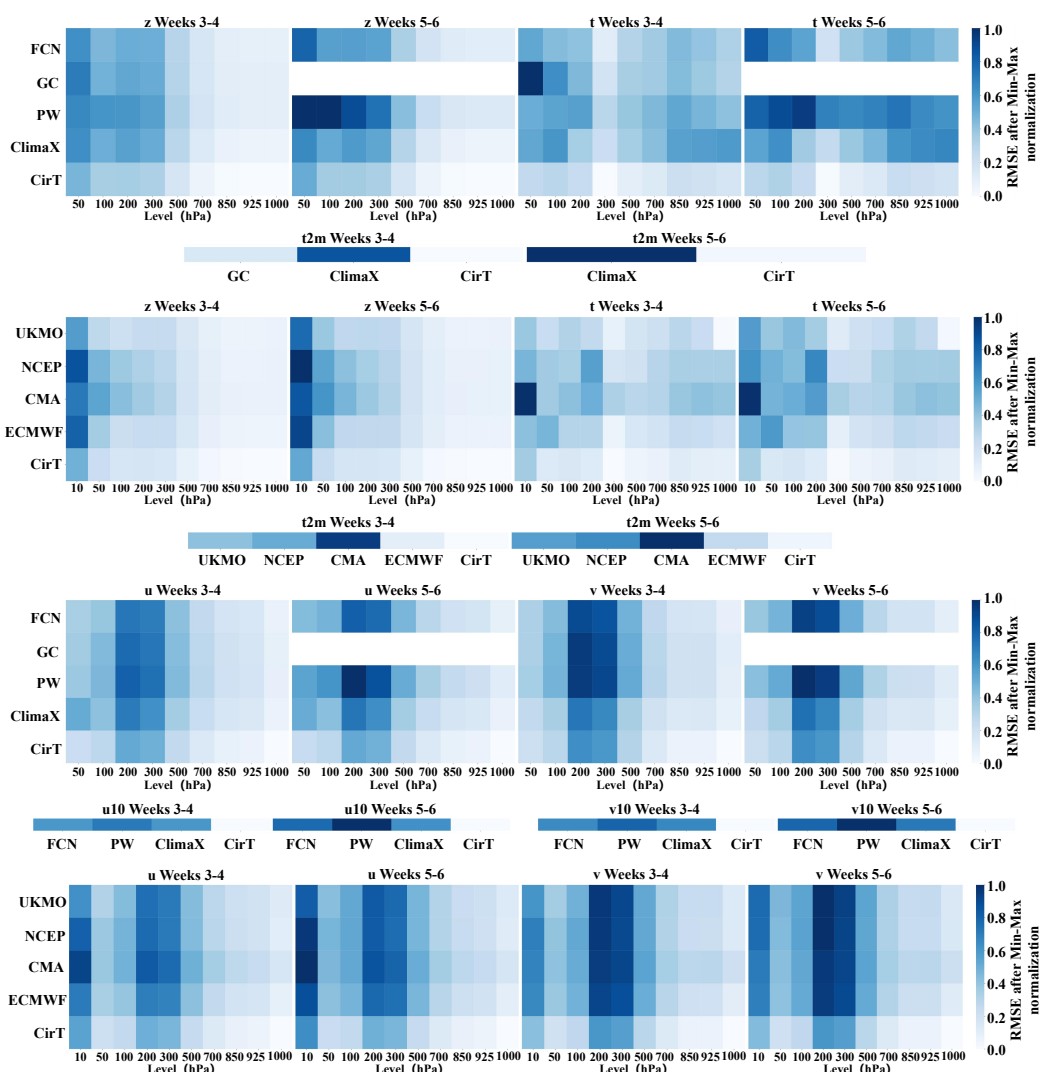

Figure 3: RMSE comparison between CirT and data-driven and numerical methods on geopotential $z$, temperature $t$, wind $u$, and $v$ of different pressure levels. FCN, GC, and PW are short for FourCastNetV2, GraphCast, and PanguWeather. A lighter color indicates better results: CirT consistently outperforms all models.

## 4.2 ABLATION STUDY

To validate the effect of each model design on the overall model performance, we compare CirT with several invariants based on whether they use grid or circular patching as well as Fourier transform. The results are shown in Table 2, where we can observe that:

- Fourier transform is a strong inductive bias and directly applying it does not always enhance model performance. From the table, we can observe that employing FT for grid patches increases the z850 and t2m errors in Weeks 3-4 as well as z850, t850, and t2m errors in Weeks 5-6, indicating the necessity of designing suitable patching approaches.

- Employing the circular patching strategy improves model performance, especially when the model uses the FT. For example, when no FT, employing circular patching reduces z500 Weeks 3-4/5-6 RMSE from 516/501 to 502/498. In contrast, in the case of the FT, employing circular patching significantly reduces z500 Weeks 3-4/5-6 RMSE from 497/494 to 477/471. Such results not only suggest the effectiveness of our model designs but also validate the effectiveness of utilizing the FT to extract the spatial periodic signal from circular patches.

Table 2: Ablation studies of patching strategies and Fourier transform.

| Patch | Fourier Transform | RMSE (↓) z500 | z850 | t500 | t850 | t2m | u10 | v10 | ACC (↑) z500 | z850 | t500 | t850 | t2m | u10 | v10 |
|---|---|---|---|---|---|---|---|---|---|---|---|---|---|---|---|
| **Weeks 3-4** | | | | | | | | | | | | | | | |
| Grid | × | 516 | 319 | 1.910 | 2.168 | 2.554 | 1.946 | 1.616 | 0.983 | 0.959 | 0.986 | 0.987 | 0.990 | 0.877 | 0.782 |
| Circular | × | 502 | 313 | 1.74 | 2.077 | 2.000 | 1.827 | 1.503 | 0.983 | 0.961 | 0.987 | 0.988 | 0.993 | 0.893 | 0.811 |
| Grid | ✓ | 497 | 324 | 1.733 | 2.050 | 2.583 | 1.970 | 1.614 | 0.983 | 0.958 | 0.988 | 0.988 | 0.990 | 0.875 | 0.782 |
| Circular | ✓ | **477** | **304** | **1.687** | **1.903** | **2.007** | **1.806** | **1.511** | **0.984** | **0.963** | **0.988** | **0.988** | **0.993** | **0.896** | **0.811** |
| **Weeks 5-6** | | | | | | | | | | | | | | | |
| Grid | × | 501 | 319 | 1.808 | 2.113 | 2.578 | 1.932 | 1.614 | 0.983 | 0.959 | 0.987 | 0.987 | 0.989 | 0.879 | 0.783 |
| Circular | × | 498 | 311 | 1.707 | 2.008 | 2.178 | 1.812 | 1.515 | 0.984 | 0.962 | 0.987 | 0.989 | 0.992 | 0.895 | 0.810 |
| Grid | ✓ | 494 | 320 | 1.737 | 2.112 | 2.650 | 1.963 | 1.621 | 0.983 | 0.960 | 0.988 | 0.988 | 0.989 | 0.877 | 0.781 |
| Circular | ✓ | **471** | **301** | **1.672** | **1.933** | **2.026** | **1.809** | **1.512** | **0.985** | **0.964** | **0.988** | **0.989** | **0.993** | **0.895** | **0.812** |

Table 3: RMSE comparison w.r.t. latitude. CirT generally achieves the best performance and has a higher relative improvement in mid-/high-latitude areas.

| | Variable | Weeks 3-4 FourCastNetV2 | GraphCast | PanguWeather | ClimaX | CirT | Weeks 5-6 FourCastNetV2 | GraphCast | PanguWeather | ClimaX | CirT |
|---|---|---|---|---|---|---|---|---|---|---|---|
| **Low-Lat.** | z500 | **200** | 229 | 293 | 234 | 206 | 223 | – | 468 | 242 | **185** |
| | z850 | 134 | 150 | 168 | 159 | **112** | 153 | – | 236 | 158 | **107** |
| | t500 | 1.090 | 1.357 | 1.490 | 1.321 | **1.076** | 1.199 | – | 2.278 | 1.364 | **1.008** |
| | t850 | 1.403 | 1.488 | 1.762 | 1.902 | **1.310** | 1.564 | – | 2.332 | 2.002 | **1.290** |
| | t2m | – | 1.363 | – | 2.044 | **1.308** | – | – | – | 2.156 | **1.264** |
| | u10 | 1.789 | – | 1.953 | 2.119 | **1.459** | 1.986 | – | 2.295 | 2.165 | **1.467** |
| | v10 | 1.399 | – | 1.494 | 1.713 | **1.165** | 1.503 | – | 1.679 | 1.779 | **1.177** |
| **Mid-Lat.** | z500 | 799 | 809 | 842 | 837 | **632** | 852 | – | 936 | 860 | **633** |
| | z850 | 521 | 539 | 541 | 506 | **404** | 552 | – | 584 | 511 | **406** |
| | t500 | 2.591 | 2.630 | 2.749 | 2.832 | **2.082** | 2.819 | – | 3.226 | 2.932 | **2.072** |
| | t850 | 2.750 | 2.787 | 2.984 | 3.093 | **2.217** | 2.950 | – | 3.349 | 3.249 | **2.260** |
| | t2m | – | 2.212 | – | 3.276 | **2.031** | – | – | – | 3.496 | **2.071** |
| | u10 | 2.895 | – | 2.967 | 2.481 | **2.154** | 3.004 | – | 3.122 | 2.474 | **2.150** |
| | v10 | 2.327 | – | 2.422 | 2.015 | **1.829** | 2.388 | – | 2.492 | 2.014 | **1.824** |
| **High-Lat.** | z500 | 1804 | 1019 | 1238 | 861 | **756** | 1856 | – | 1299 | 880 | **743** |
| | z850 | 1050 | 689 | 715 | 566 | **514** | 1074 | – | 772 | 569 | **498** |
| | t500 | 5.249 | 3.297 | 4.667 | 2.833 | **2.414** | 5.465 | – | 4.857 | 2.879 | **2.497** |
| | t850 | 7.927 | 3.832 | 6.265 | 3.560 | **2.833** | 8.244 | – | 6.459 | 3.678 | **2.955** |
| | t2m | – | 4.153 | – | 4.874 | **3.601** | – | – | – | 5.086 | **3.674** |
| | u10 | 2.559 | – | 2.567 | 2.641 | **1.918** | 2.414 | – | 2.453 | 2.377 | **1.765** |
| | v10 | 2.625 | – | 2.661 | 2.657 | **1.927** | 2.457 | – | 2.493 | 2.392 | **1.755** |

## 4.3 EMPIRICAL ANALYSIS

**Latitudinal forecasting** To investigate the model performance w.r.t. the latitude, we compare their results at low-latitude (0°-30°), mid-latitude (30°-60°), and high-latitude (60°-90°) areas. The results are displayed in Table 3. From them, we can discover that CirT generally outperforms baselines in all areas. Moreover, CirT achieves larger relative improvement in mid-latitude and high-latitude areas. For example, in the t500 Weeks 3-4 prediction, CirT has a 1.2% relative improvement in low-latitude areas over the best baseline and 19.6%/16.2% in mid-/high-latitude areas. Such improvement can be attributed to the consideration of geometric inductive bias.

**Global visualization** To provide a global view of model predictions, we visualize the Weeks 3-4 RMSE distribution in Figure 4. More visualizations such as Weeks 5-6 predictions and different variables can be found in the Appendix. As shown in the figure, all models achieve the best results in the equatorial region. Higher RMSE values are primarily concentrated in south polar areas and the continents of the Northern Hemisphere, especially in North America and the Bering Strait. Among these methods, CirT demonstrates significantly lower errors, even in the above areas, further validating our framework of incorporating geometric information.

**Performance w.r.t. month** We further compare the model performance over different months, which is shown in Figure 5. The results of other variables can be found in the Appendix. CirT has superior predictive capabilities in forecasting at all times, outperforming all competing methods

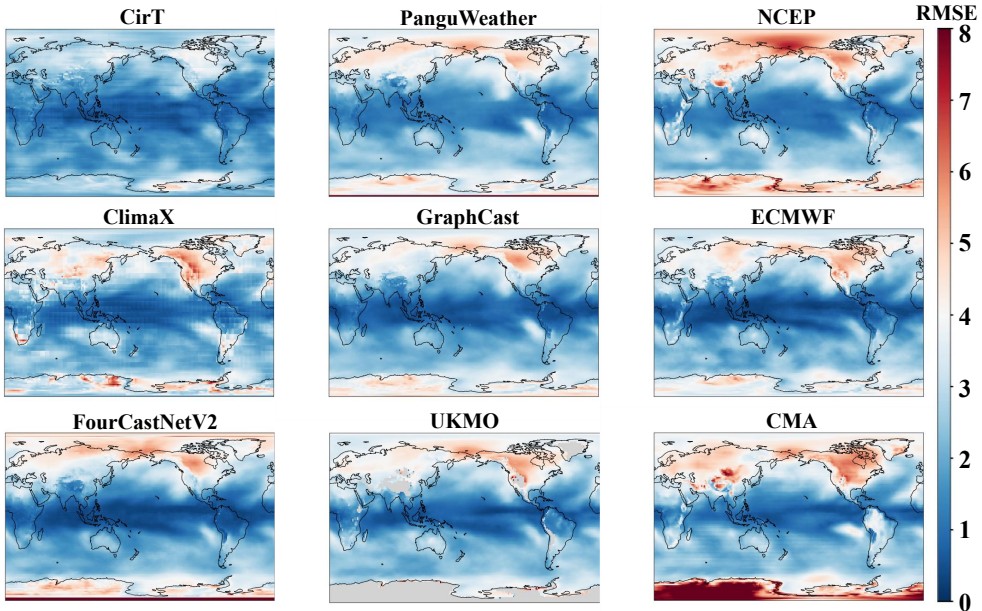

Figure 4: The global RMSE distribution of t850 with lead times weeks 3-4 in testing set: CirT demonstrates significant performance across different areas.

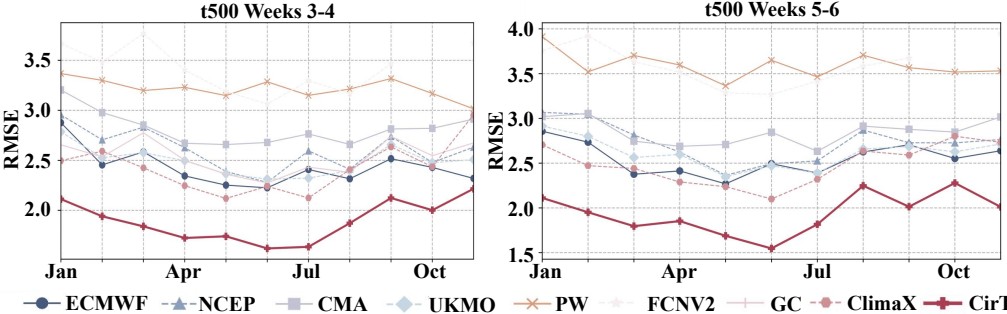

Figure 5: The monthly RMSE of t500 in testing set: CirT outperforms baselines across all months.

at all months. In particular, it has the largest improvement in June for both Weeks 3-4 and 5-6 predictions.

## 5 RELATED WORK

The advances in numerical weather prediction have dominated weather and climate modeling over the last century. They model the complex Earth dynamics as coupled physical systems such as earth system models (ESM) (Hurrell et al., 2013), integrating the simulations of the atmosphere, cryosphere, land, and ocean processes. With the development of machine learning models (Ho et al., 2019; Guibas et al., 2021) and the accessibility of high-quality weather data, various data-driven models have been proposed to mitigate the weaknesses of NWP such as high computational demands and sensitivity to initial conditions. Early studies target regional forecasting on specific variables, such as precipitation nowcasting in Hong Kong (Shi et al., 2015), wind prediction in Stuttgart (Harbola & Coors, 2019), and air temperature prediction in Australia (Deo, 2016). A notable progress is the publication of the ECMWF reanalysis v5 (ERA5) dataset (Hersbach et al., 2020), which combines historical observations with results from a high-fidelity integrated Forecasting System (IFS) (Wedi et al., 2015). Based on such a dataset, pioneer works (Scher, 2018; Weyn et al., 2019; 2020; Rasp et al., 2020; Rasp & Thuerey, 2021) study the global forecasting of specific

variables such as 500 hPa Geopotential and 300 hPa zonal wind, but at relatively coarser resolutions (Verma et al., 2024).

Recently, the development of foundation models has significantly advanced data-driven weather and climate forecasting. They are trained on large-scale high-resolution global data and target various weather variables. FourCastNet (Pathak et al., 2022) and a follow-up work SFNO (Bonev et al., 2023) are built on the framework of the Fourier Neural Operator (Li et al., 2020; Guibas et al., 2021). Graph-based models (Lam et al., 2022; Keisler, 2022) such as GraphCast (Keisler, 2022) first create a mesh grid on the spherical surface and perform message passing (Liu et al., 2024; Zheng et al., 2024; Liu et al., 2022) on it. Besides these studies, a series of works are based on Transformer (Vaswani, 2017; Dosovitskiy, 2020). Pangu-Weather (Bi et al., 2023) leverages sliding window attention to model spatial relations. Based on a similar backbone, Fuxi (Chen et al., 2023b) and FengWu (Chen et al., 2023a) improve training strategies to reduce accumulation errors and incorporate multi-model/task perspectives respectively. Climax (Nguyen et al., 2023) demonstrates its ability for various weather and climate tasks and CaFA (Li et al., 2024) considers the spherical geometry.

Despite the progress of current foundation models, S2S forecasting receives less attention due to its difficulty. Hwang et al. (Hwang et al., 2019) and He et al. (He et al., 2022) study regional S2S forecasting via traditional machine learning models such as AutoKNN and XGBoost (Chen & Guestrin, 2016). Weyn et al. (Weyn et al., 2021) designs an ensemble system based on convolution neural networks to predict six atmospheric variables. It's only been recently that Climax (Nguyen et al., 2023) and Fuxi-S2S (Chen et al., 2024) have been developed to try to tackle these issues based on pre-trained foundation models. Therefore, how to build an effective data-driven S2S forecasting model is still an open problem. In this work, we propose CirT and study the performance of the direct prediction model and show that it outperforms current iterative models. Moreover, existing S2S models generally treat global data as planar which introduces geometric inconsistency while we leverage spherical inductive bias in model designs to alleviate such problems.

Although both GraphCast (Lam et al., 2022) and CirT leverage geometric inductive biases, Graph-Cast focuses on local state aggregation and relies on message passing (Liu et al., 2023) to aggregate local information without explicitly accounting for spatial periodicity. In contrast, CirT employs circular patching to normalize patch geometry and leverages its Fourier representation, consisting of coefficients of periodic basis functions, as inputs to the transformer encoders. Compared with Four-castNet (Pathak et al., 2022) which aims to design an efficient token mixer for Vision Transformers that can effectively handle high-resolution inputs, CirT performs multi-head attention in the frequency domain to model the interactions among weather patches across various latitudes. Moreover, FourcastNet employs regular grid patching while CirT introduces circular patching to standardize patch geometry. Pangu Weather also designs an earth-specific positional bias to integrate spherical information, encoding relative coordinates with learnable parameters into the attention weight computation. Nevertheless, they still employ cube patching and implicitly learn the geometric bias from data, while CirT explicitly leverages spherical bias in the patching strategy.

## 6 CONCLUSION AND FUTURE WORK

In this work, we highlight the geometric inductive bias in Transformer designs for S2S forecasting and introduce CirT, consisting of a circular patching strategy and latitudinal spatial periodicity modeling. It learns to mix patch embeddings in frequency domains and inverse transform to the spatial domain. Finally, it is trained to predict the future states in the S2S timescale. Extensive experiments on the ERA5 dataset demonstrate that CirT not only outperforms advanced data-driven models but also skillful numerical methods. Ablation studies have further substantiated the effectiveness of model designs and additional empirical analysis illustrates the superior performance in spatial and time dimensions. In the future, we are interested in incorporating slowly evolving earth system components including ocean, land, and sea ice in the proposed framework. In addition, CirT is a 2D transformer that encodes the inputs of different pressure levels into a single embedding. Nevertheless, failure to incorporate the vertical inductive biases may result in incomplete cross-pressure level interaction. We plan to generalize the 2D CirT to the 3D transformer to incorporate such inductive biases.

ACKNOWLEDGMENTS

This work is funded by National Natural Science Foundation of China Grant No. 72371217, the Guangzhou Industrial Informatic and Intelligence Key Laboratory No. 2024A03J0628, the Nansha Key Area Science and Technology Project No. 2023ZD003, and Project No. 2021JC02X191.

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

# A APPENDIX

## A.1 ADDITIONAL EXPERIMENTAL DETAILS

### A.1.1 BASELINE

- **UKMO** The UK Meteorological Office uses the Global Seasonal forecast system version 6 (GloSea6) model to generate daily control forecasts for 60-day lead time.

- **NCEP** The National Centers for Environmental Prediction uses the Climate Forecast System 2 (CFSv2) model to generate daily control forecasts for 45-day lead time.

- **CMA** The China Meteorological Administration uses the Beijing Climate Center (BCC) fully-coupled BCC-CSM2-HR model to generate control forecasts each Monday and Thursday for 60-day lead time.

- **ECMWF** The European Centre for Medium-Range Weather Forecasts uses the operational Integrated Forecasting System (IFS) to generate control forecasts each Monday and Thursday for 46-day lead time. We use the CY48R1 version to forcast.

- **FourCastNetV2** Iterative data-driven model built upon Vision Transformer. FourCastNetV2 patches all the variables and uses the Adaptive Fourier Neural Operator to mix the spatial patches. We use the API https://github.com/ecmwf-lab/ai-models to perform inference. Due to the data loss in October 2018, we utilized the available data from the remaining 11 months.

- **GraphCast** Iterative data-driven model built upon Graph Neural Network. It use multi-mesh method to construct the graph to learn the complex dynamics. We use the API https://github.com/ecmwf-lab/ai-models to perform inference. The GPU's memory only allows us to perform inference on data spanning a maximum of 4 weeks. Therefore, we only present the results for weeks 3-4.

- **PanguWeather** Iterative data-driven model built upon Vision Transformer. We use the API https://github.com/ecmwf-lab/ai-models to perform inference. PanguWeather patches the pressure level and single-level data separately and merge them in the transformer. Using hierarchical temporal aggregation method to train the mdoel.

- **ClimaX** Direct training data-driven model built upon Vision Transformer. Each variable is independently tokenized and aggregated by variable aggregation.

### A.1.2 DATASET

For all data-driven models, we utilize the same approach to transform $0.25° \times 0.25°$ grid to $1.5° \times 1.5°$ grid, including obtaining the prediction of FourCastNetV2, Graphcast, and PanguWeather as well as the training grid data of CirT. Specifically, we first obtain the results of $0.25° \times 0.25°$ models (e.g., PanguWeather), which are represented on a $721 \times 1440$ grid. This grid corresponds to the coordinates $(\lambda, \phi)$ within the domain $\Omega = [-90°, -89.75°, \ldots, 89.75°, 90°] \times [-180°, -179.75°, \ldots, 179.75°, 180°]$, where $\lambda$ denotes longitude and $\phi$ denotes latitude. Subsequently, we retrieve the results of coordinates $(\lambda, \phi)$ that correspond to the $1.5° \times 1.5°$ grid, which is represented on a $121 \times 240$ grid within the domain $\Omega = [-90°, -88.5°, \ldots, 88.5°, 90°] \times [-180°, -178.5°, \ldots, 178.5°, 180°]$.

## A.2 ADDITIONAL RESULTS

**Acc results** ACC comparison between CirT and numerical models at all pressure levels are shown in Figure 6 and ACC comparison w.r.t. latitudes are displayed in Table 9.

**Regional forecasting** We additionally evaluate models in regional forecasting, constrained to the bounding boxes of North America and Europe. The results are shown in Table 7. We can observe that CirT outperforms baselines in all cases.

**Additional visualization** The global visualization of t850 Weeks 5-6 predictions and other variables including t500, z500, and z850 are shown in Figure 7-13.

Table 4: Computation complexity and model size comparison.

| Model | GraphCast | PanguWeather | CirT |
|---|---|---|---|
| FLOPs | 110 T | 168 T | 2.2 G |
| Params | 37M | 256M | 16M |

Table 5: Ablation study of autoregressive prediction vs directly predicting all the future values.

| Model | | RMSE (↓) | | | | | | ACC (↑) | | | | | | |
|---|---|---|---|---|---|---|---|---|---|---|---|---|---|---|
| | z500 | z850 | t500 | t850 | t2m | u10 | v10 | z500 | z850 | t500 | t850 | t2m | u10 | v10 |
| **3-4** Autoregressive | 781 | 453 | 3.406 | 4.014 | 4.584 | 2.806 | 2.267 | 0.962 | 0.922 | 0.956 | 0.957 | 0.968 | 0.763 | 0.610 |
| Direct | **477** | **304** | **1.687** | **1.903** | **2.007** | **1.806** | **1.511** | **0.984** | **0.963** | **0.988** | **0.988** | **0.993** | **0.896** | **0.811** |
| **5-6** Autoregressive | 813 | 455 | 3.636 | 4.357 | 5.047 | 2.855 | 2.324 | 0.960 | 0.923 | 0.950 | 0.949 | 0.960 | 0.758 | 0.599 |
| Direct | **471** | **301** | **1.672** | **1.933** | **2.026** | **1.809** | **1.512** | **0.985** | **0.964** | **0.988** | **0.989** | **0.993** | **0.895** | **0.812** |

**Additional results w.r.t month**    Additional results of variables t850, z500, and z850 are shown in Figure 14-16.

**Comparison of model computation complexity/size**    We compare CirT's Floating point operations (FLOPs) and parameters with two representative models, Graphcast and PanguWeather. The results are shown in Table 4. We can observe that CirT achieves better S2S predictivity with less computation and smaller model size, verifying our model designs.

**Comparison of autoregressive and direct prediction**    We adapted CirT's output head to forecast next-day weather variables based on the input date for autoregressive prediction. For inference, it iteratively predicts next-day weather variables up to the S2S timescale. The results are shown in Table 5. From the results, we can observe that the direct method performs better. The autoregressive CirT still accumulates errors, resulting in inaccurate S2S predictions.

**Additional results on fine-tuning CirT.**    We further evaluate the performance of fine-tuning the trained autoregressive CirT. We freeze the transformer encoder and replace the embedding layers and output head with newly initialized networks to forecast weather variables for Weeks 3-4 and 5-6. The results are in Table 6. From the result, we can observe that direct training still performs best in most cases. Meanwhile, we find that fine-tuned embedding layer and decoder improve the performance in several variables such as $t850$.

**Multi-scale structural similarity**    Following the previous work (Nathaniel et al., 2024), we also compare the Multi-Scale Structural Similarity (Wang et al., 2003) of the data-driven models. The result are shown in Table 8. CirT achieves the best performance. GraphCast is the best baseline for Weeks 3-4 predictions. The reason can be attributed to that it employs mesh to model the sphere geometry, consistent with the observations in Table 1.

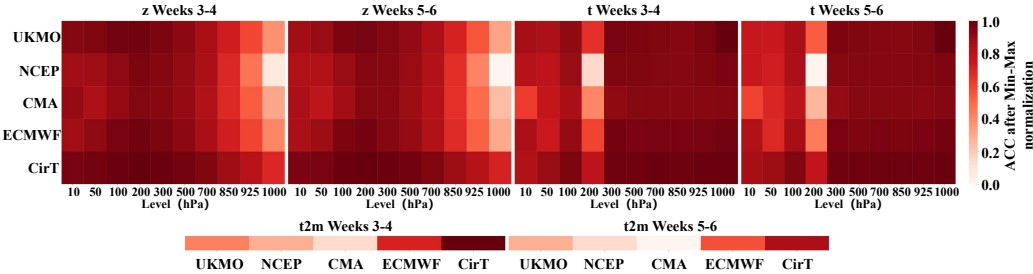

Figure 6: The ACC comparison between numerical models and CirT.

Table 6: Ablation study of fine-tuning CirT model.

| | Model | RMSE (↓) | | | | | | |
| | | z500 | z850 | t500 | t850 | t2m | u10 | v10 |
|---|---|---|---|---|---|---|---|---|
| **3-4** | Fine-tuning embedding and decoder | 480 | 315 | **1.660** | **1.870** | **1.983** | 1.842 | 1.530 |
| | Fine-tuning Decoder | 540 | 346 | 1.885 | 2.327 | 2.715 | 2.013 | 1.619 |
| | Direct Training | **477** | **304** | 1.687 | 1.903 | 2.007 | **1.806** | **1.511** |
| **5-6** | Fine-tuning embedding and decoder | 485 | 312 | 1.679 | **1.923** | 2.032 | 1.847 | 1.535 |
| | Fine-tuning Decoder | 588 | 354 | 2.190 | 2.702 | 3.145 | 2.043 | 1.650 |
| | Direct Training | **471** | **301** | **1.672** | 1.933 | **2.026** | **1.809** | **1.512** |

Table 7: RMSE comparison w.r.t. regions. CirT generally achieves the best performance and has a higher relative improvement in mid-/high-latitude areas. N-America is short for North America

| | Variable | Weeks 3-4 | | | | | Week 5-6 | | | | |
| | | FourCastNetV2 | GraphCast | PanguWeather | ClimaX | CirT | FourCastNetV2 | GraphCast | PanguWeather | ClimaX | CirT |
|---|---|---|---|---|---|---|---|---|---|---|---|
| **N-America** | z500 | 1017 | 1014 | 996 | 879 | **801** | 1037 | – | 1063 | 916 | **774** |
| | z850 | 709 | 714 | 690 | 606 | **552** | 715 | – | 744 | 634 | **538** |
| | t500 | 2.770 | 2.704 | 2.793 | 2.392 | **2.128** | 2.858 | – | 3.221 | 2.431 | **2.136** |
| | t850 | 3.012 | 2.962 | 3.041 | 2.802 | **2.333** | 3.110 | – | 3.454 | 2.826 | **2.347** |
| | t2m | – | 3.184 | – | 3.618 | **2.556** | – | – | – | 3.669 | **2.617** |
| **Europe** | z500 | 892 | 905 | 909 | 855 | **651** | 953 | – | 995 | 854 | **640** |
| | z850 | 606 | 617 | 620 | 551 | **427** | 636 | – | 675 | 555 | **415** |
| | t500 | 2.862 | 2.828 | 2.869 | 2.573 | **2.111** | 3.094 | – | 3.282 | 2.516 | **2.163** |
| | t850 | 2.848 | 2.910 | 3.035 | 2.827 | **2.083** | 3.078 | – | 3.382 | 2.828 | **2.139** |
| | t2m | – | 2.972 | – | 3.561 | **2.334** | – | – | – | 3.619 | **2.396** |

Table 8: Multi-scale structural similarity of data-driven models.

| Variable | Week 3-4 | | | | | Week 5-6 | | | | |
| | FourCastNetV2 | GraphCast | PanguWeather | ClimaX | CirT | FourCastNetV2 | GraphCast | PanguWeather | ClimaX | CirT |
|---|---|---|---|---|---|---|---|---|---|---|
| z500 | 0.814 | 0.872 | 0.865 | 0.862 | **0.909** | 0.808 | – | 0.846 | 0.854 | **0.909** |
| z850 | 0.799 | 0.811 | 0.802 | 0.794 | **0.874** | 0.786 | – | 0.772 | 0.789 | **0.874** |
| t500 | 0.866 | 0.889 | 0.882 | 0.875 | **0.925** | 0.857 | – | 0.860 | 0.869 | **0.924** |
| t850 | 0.882 | 0.919 | 0.913 | 0.893 | **0.942** | 0.876 | – | 0.901 | 0.885 | **0.942** |
| t2m | – | 0.966 | – | 0.928 | **0.969** | – | – | – | 0.921 | **0.968** |

Table 9: ACC comparison w.r.t. latitude. CirT generally achieves the best performance and has a higher relative improvement in mid-/high-latitude areas.

| | Variable | Weeks 3-4 | | | | | Week 5-6 | | | | |
| | | FourCastNetV2 | GraphCast | PanguWeather | ClimaX | CirT | FourCastNetV2 | GraphCast | PanguWeather | ClimaX | CirT |
|---|---|---|---|---|---|---|---|---|---|---|---|
| **Low-Lat.** | z500 | 0.998 | 0.998 | 0.998 | 0.997 | **0.999** | 0.998 | – | 0.997 | 0.997 | **0.999** |
| | z850 | 0.993 | 0.991 | 0.990 | 0.991 | **0.995** | 0.991 | – | 0.983 | 0.991 | **0.996** |
| | t500 | 0.997 | 0.996 | 0.996 | 0.995 | **0.998** | 0.996 | – | 0.994 | 0.995 | **0.997** |
| | t850 | 0.995 | 0.995 | 0.995 | 0.993 | **0.997** | 0.995 | – | 0.993 | 0.992 | **0.996** |
| | t2m | – | 0.997 | – | 0.994 | **0.998** | – | – | – | 0.994 | **0.998** |
| **Mid-Lat.** | z500 | 0.932 | 0.927 | 0.921 | 0.920 | **0.955** | 0.923 | – | 0.902 | 0.915 | **0.954** |
| | z850 | 0.888 | 0.878 | 0.874 | 0.887 | **0.929** | 0.876 | – | 0.852 | 0.886 | **0.929** |
| | t500 | 0.943 | 0.937 | 0.932 | 0.923 | **0.959** | 0.932 | – | 0.907 | 0.917 | **0.959** |
| | t850 | 0.951 | 0.947 | 0.939 | 0.931 | **0.965** | 0.942 | – | 0.922 | 0.923 | **0.963** |
| | t2m | – | 0.976 | – | 0.944 | **0.978** | – | – | – | 0.936 | **0.978** |
| **High-Lat.** | z500 | 0.927 | 0.969 | 0.952 | 0.978 | **0.983** | 0.923 | – | 0.948 | 0.977 | **0.984** |
| | z850 | 0.870 | 0.925 | 0.918 | 0.952 | **0.961** | 0.863 | – | 0.904 | 0.951 | **0.963** |
| | t500 | 0.951 | 0.978 | 0.952 | 0.982 | **0.987** | 0.948 | – | 0.952 | 0.981 | **0.988** |
| | t850 | 0.942 | 0.982 | 0.946 | 0.983 | **0.989** | 0.938 | – | 0.944 | 0.982 | **0.989** |
| | t2m | – | 0.990 | – | 0.986 | **0.992** | – | – | – | 0.984 | **0.992** |

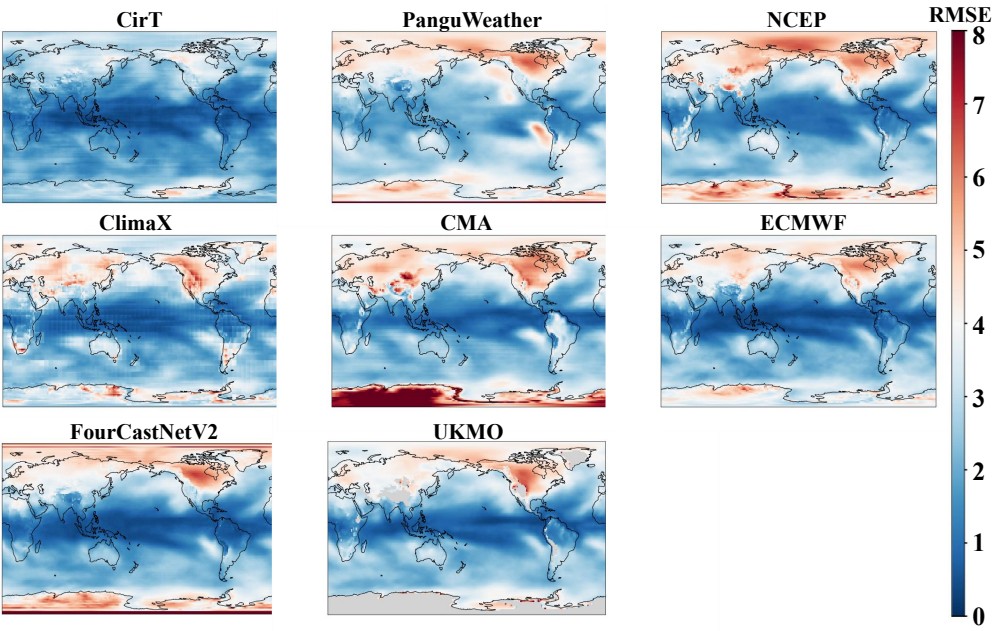

Figure 7: The global RMSE maps of t850 with lead times weeks 5-6 in 2018.

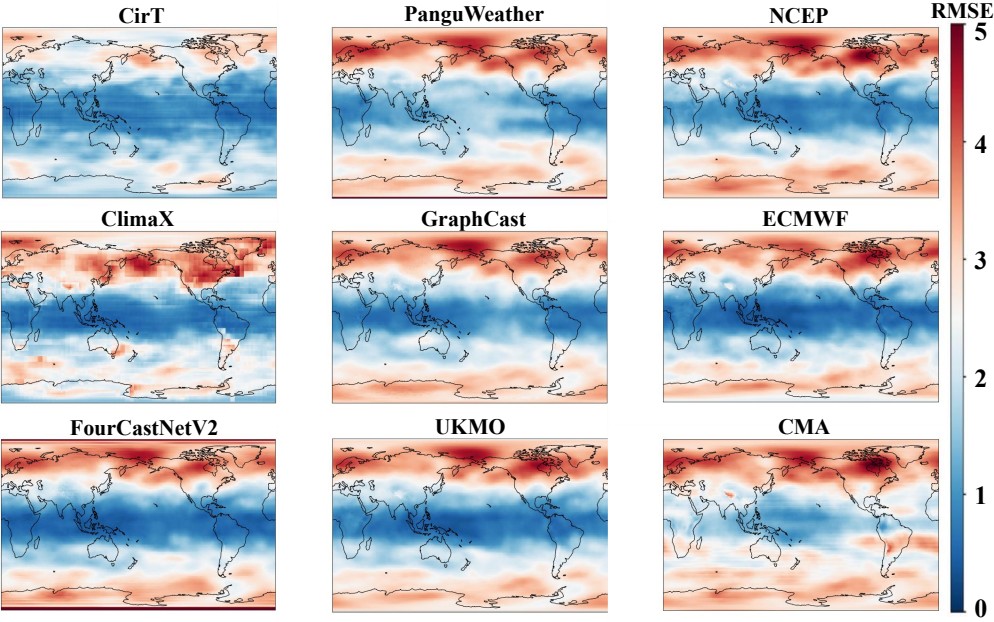

Figure 8: The global RMSE maps of t500 with lead times weeks 3-4 in 2018.

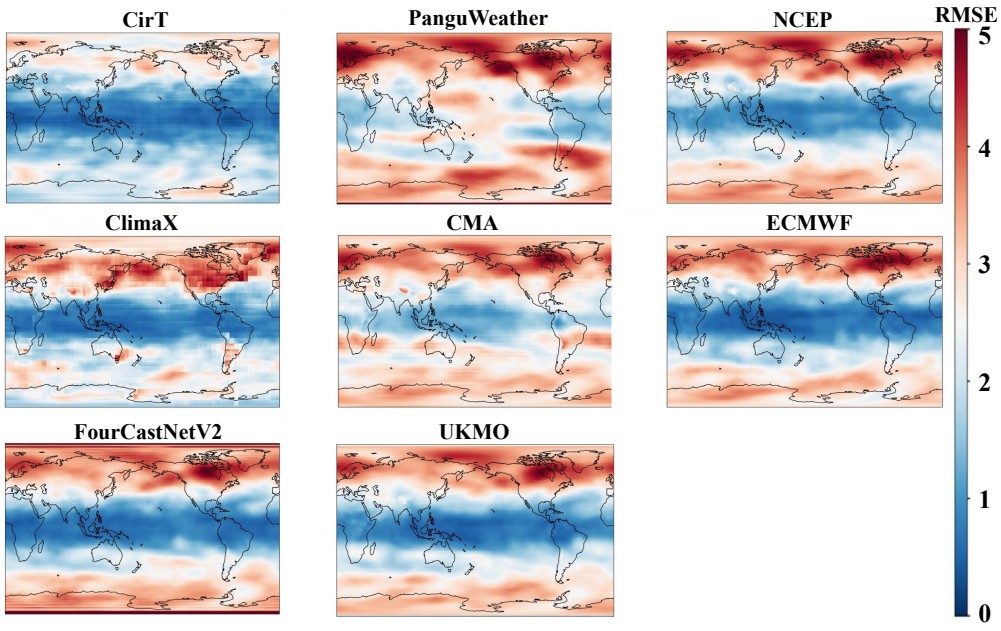

Figure 9: The global RMSE maps of t500 with lead times weeks 5-6 in 2018.

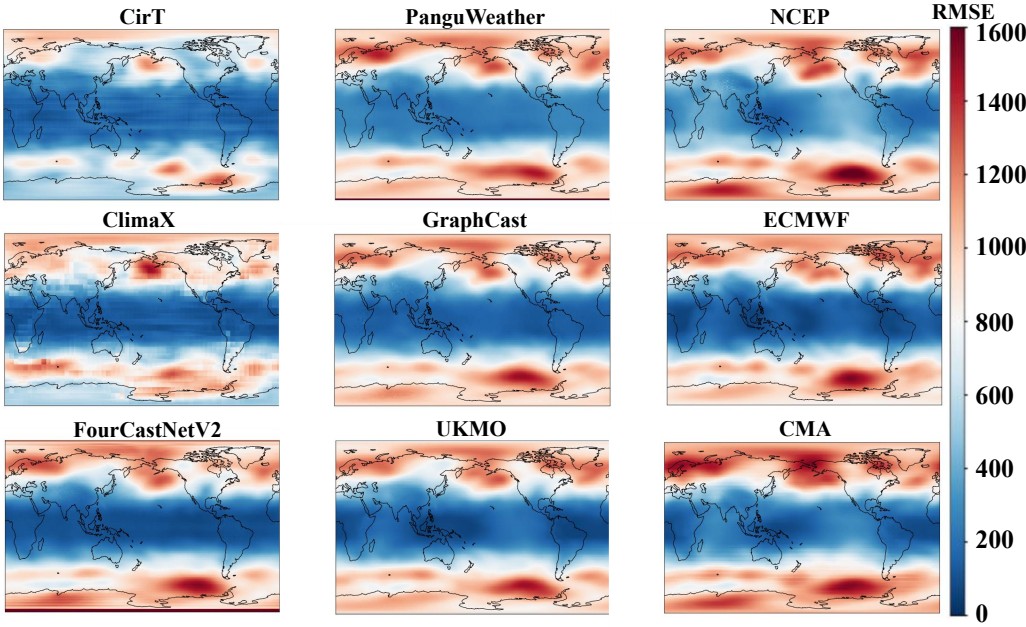

Figure 10: The global RMSE maps of z500 with lead times weeks 3-4 in 2018.

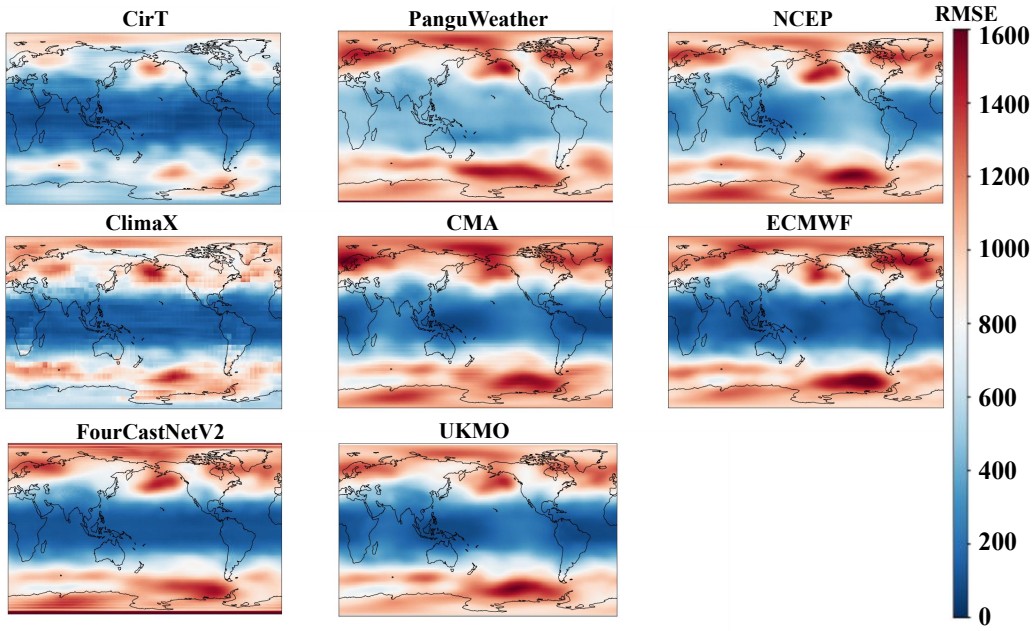

Figure 11: The global RMSE maps of z500 with lead times weeks 5-6 in 2018.

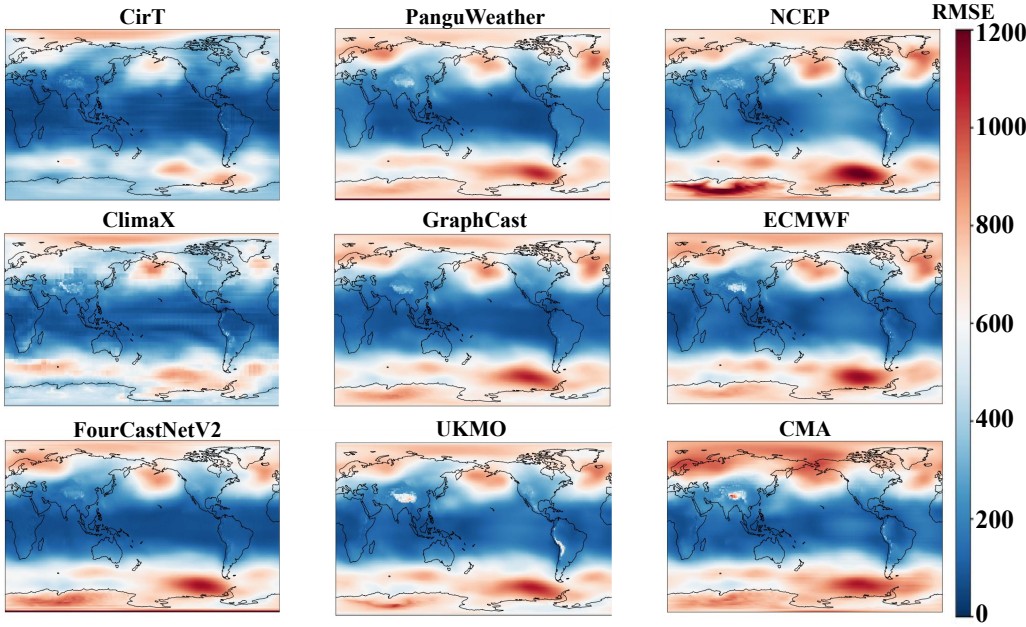

Figure 12: The global RMSE maps of z850 with lead times weeks 3-4 in 2018.

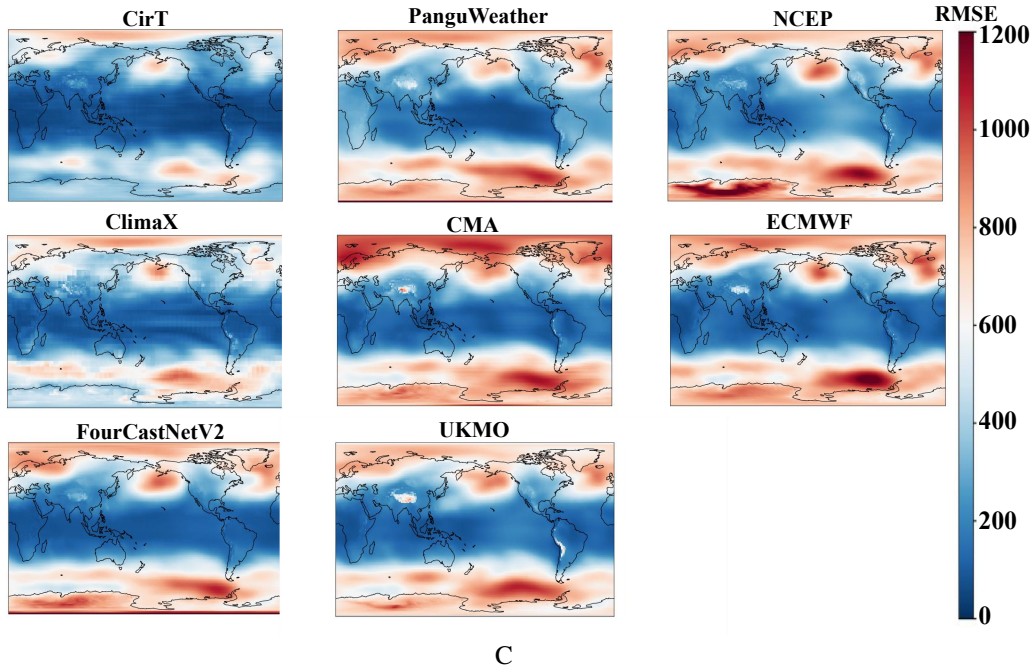

Figure 13: The global RMSE maps of z850 with lead times weeks 5-6 in 2018.

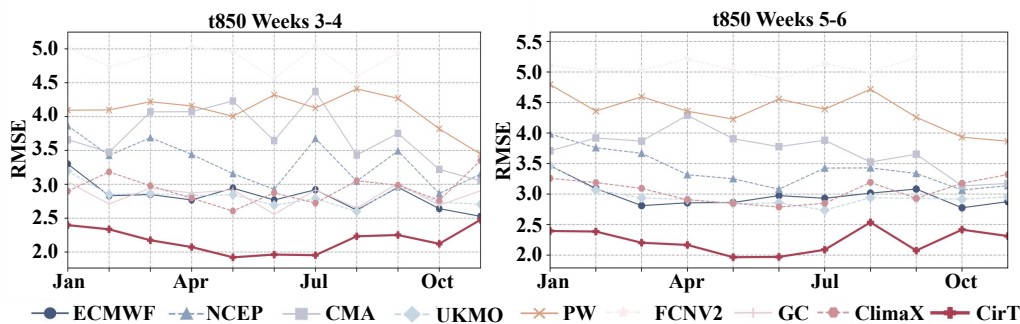

Figure 14: The monthly RMSE of t850 in testing set: CirT outperforms other models across all months.

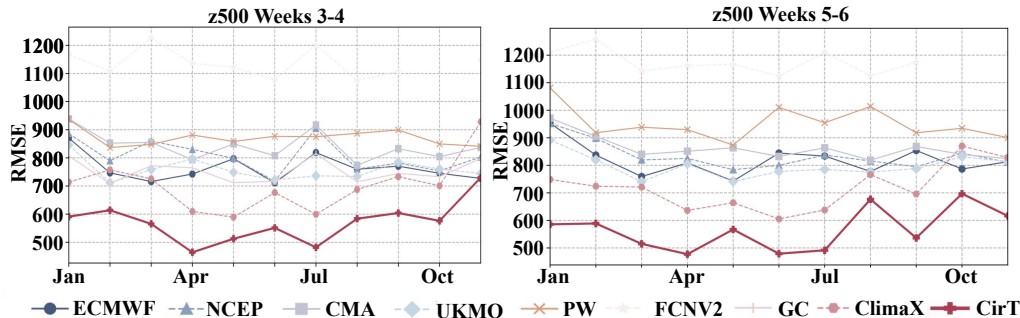

Figure 15: The monthly RMSE of z500 in testing set: CirT outperforms other models across all months.

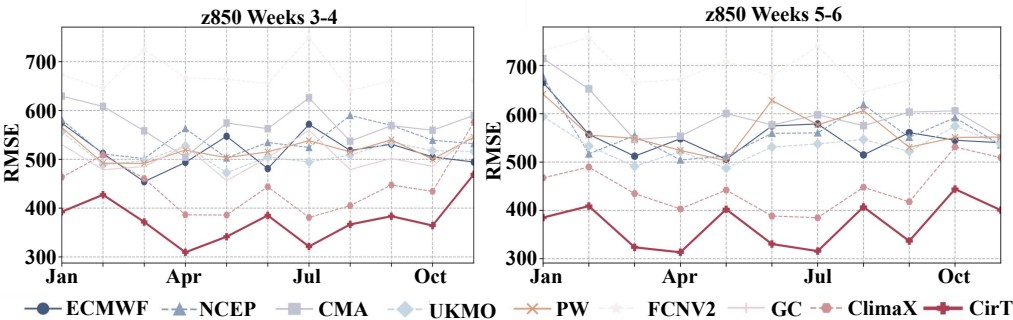

Figure 16: The monthly RMSE of z850 in testing set: CirT outperforms other models across all months.

