# OpenReview forum: "CirT: Global Subseasonal-to-Seasonal Forecasting with Geometry-inspired Transformer"
_ICLR.cc/2025/Conference — ICLR 2025 Poster_

### Official Review · Reviewer_iiXK · 2024-10-31

**Soundness:** 2
**Presentation:** 3
**Contribution:** 2
**Rating:** 6
**Confidence:** 5

**Summary:**

The article designs a circular transformer (CirT) to model medium—and long-term numerical forecasts. It takes into account the spherical structure of the ground, designs corresponding circular patches as input tokens of the Transformer, and uses the Fourier transform in self-attention to capture global information to model the spatial periodicity, achieving high accuracy.

**Strengths:**

The article has clear logic and ideas. The proposed method is very consistent with the research problem. It considers the problems faced by Patch Embedding in global meteorological modeling. Compared with most methods, it has achieved excellent results in high-latitude regions.

**Weaknesses:**

1、The data in the experiment of the article is 1.5° instead of 0.25° as used in the Pangu model. Obviously, the author should give a detailed explanation of the experiment.
2、The article lacks experimental conditions for key variables such as wind u and wind v.
3、The article states that the usage of the Transformer network with Fourier transform is an important component, but both AFNO and Fourcast use related structures, which is not innovative enough.

**Questions:**

1、The author should explain in detail how the results of the 1.5° experiment of the Pangu model were obtained. In global forecasts, there is currently a 0.25° prediction model. Is the 1.5° model necessary?
2、The experimental conditions of variables such as wind u and wind v.
3、Most weather models are tuned after training is complete, the article does not perform this work and fine-tuning can be used to further improve model performance.

---

> ### Author Response · Authors · 2024-11-19
> **Author Responses (1/2)**
>
> We sincerely thank you for recognizing the idea and results of our work. We have made every effort to faithfully address your comments in the following responses.
>
> >Weakness 1 & Questions 1: A detailed explanation of the $1.5^{\circ}$ experimental results of $0.25^{\circ}$ models.
>
> Thanks for raising this concern. We compare model results on latitude-longitude grid points. For all models, we utilize the same approach to transform $0.25^{\circ}\times 0.25^{\circ}$ grid to $1.5^{\circ}\times 1.5^{\circ}$ grid, including obtaining the prediction of FourCastNetV2, Graphcast, and PanguWeather as well as the training grid data of CirT. Specifically, we first obtain the results of $0.25^{\circ}\times 0.25^{\circ}$ models (e.g., PanguWeather), which are represented on a $721 \times 1440$ grid. This grid corresponds to the coordinates $(\lambda, \phi)$ within the domain $\Omega = [-90^{\circ}, -89.75^{\circ}, \ldots, 89.75^{\circ}, 90^{\circ}] \times [-180^{\circ}, -179.75^{\circ}, \ldots, 179.75^{\circ}, 180^{\circ}]$, where $\lambda$ denotes longitude and $\phi$ denotes latitude. Subsequently, we retrieve the results of coordinates $(\lambda, \phi)$ that correspond to the $1.5^{\circ}\times 1.5^{\circ}$ grid, which is represented on a $121 \times 240$ grid within the domain $\Omega = [-90^{\circ}, -88.5^{\circ}, \ldots, 88.5^{\circ}, 90^{\circ}] \times [-180^{\circ}, -178.5^{\circ}, \ldots, 178.5^{\circ}, 180^{\circ}]$.
>
> We can find that CirT outperforms $0.25^{\circ}\times 0.25^{\circ}$ models even is trained on lower-resolution data, verifying its effectiveness. Please see our response to the subsequent question explaining the rationale for choosing a $1.5^{\circ}\times 1.5^{\circ}$ resolution.
>
> >Questions 1: In global forecasts, there is currently a $0.25^{\circ}$ prediction model. Is the $1.5^{\circ}$ model necessary?
>
> Thanks for the question. We would like to clarify that global medium-range forecasting and S2S forecasting are distinct. The existing $0.25^{\circ}\times 0.25^{\circ}$ model focuses on medium-range (up to 15 days) forecasting while S2S forecasting generally considers coarse-grained resolutions like $1.5^{\circ}\times 1.5^{\circ}$. Specifically,
>
> * Predictions on the S2S timescale are highly challenging and have long been considered a “predictability desert”. Therefore, to enhance predictability, most methods[1-2] consider a coarse-grained $1.5^{\circ}\times 1.5^{\circ}$ resolution.
> * As shown in Table 1 and Figure 3, although $0.25^{\circ}\times 0.25^{\circ}$ model (e.g., PanguWeather) are trained on higher-resolution data, their S2S predictability does not demonstrate superior performance and underform CirT.
>
> Therefore, $1.5^{\circ}$ S2S prediction model is still necessary.
>
> [1] A machine learning model that outperforms conventional global subseasonal forecast models, Nature Communications, 2024.
> [2] ChaosBench: a multi-channel, physics-based benchmark for subseasonal-to-seasonal climate prediction, NeurIPS 2024.
>
>
> >Weakness 2 & Questions 2: The article lacks experimental conditions for key variables such as wind u and wind v.
>
> Thanks for your constructive comment. We add model comparisons of 10 metre U wind component ($u10$) and 10 metre V wind component ($v10$) in Table 1 of the revised PDF. Note that the retrieved GraphCast model in ECMWF does not include their inference. The results are shown as follows:
>
> Results of Weeks 3-4 prediction:
> | RMSE | FourCastNetV2 |  PanguWeather | Climax | CirT |
> | :----|:----|:----|:----|:----|
> | $u10$ | 2.328 | 2.431 | 2.334 | 1.806 |
> | $v10$ | 1.896 | 1.984 | 1.906 | 1.511 |
>
> | ACC | FourCastNetV2 |  PanguWeather | Climax | CirT |
> | :----|:----|:----|:----|:----|
> | $u10$ | 0.830 | 0.812 | 0.817 | 0.896 |
> | $v10$ | 0.712  | 0.686 | 0.667 | 0.811 |
>
> Results of Weeks 5-6 prediction:
> | RMSE | FourCastNetV2 |  PanguWeather | Climax | CirT |
> | :----|:----|:----|:----|:----|
> | $u10$ | 2.479  | 2.679 | 2.355 | 1.809 |
> | $v10$ | 1.980  | 2.104 | 1.939 | 1.512 |
>
> | ACC | FourCastNetV2 |  PanguWeather | Climax | CirT |
> | :----|:----|:----|:----|:----|
> | $u10$ | 0.812  | 0.783 | 0.814 | 0.895 |
> | $v10$ | 0.691  | 0.655 | 0.659 | 0.812 |
>
> We can find that wind forecasting is more challenging than other comparing variables. For example, Weeks 3-4 t850 ACC of FourCastNetV2 is 0.957 while u10 ACC is 0.830. Under such cases, CirT still performs the best, further verifying its effectiveness.
>
> In addition, we add the following results regarding the wind $u$ and $v$:
>
> * RMSE comparison with data-driven methods w.r.t. different pressure levels of $u$ and $v$ (see Figure 3 of the revised PDF).
> * RMSE comparison with numerical methods w.r.t. different pressure levels of $u$ and $v$. (see Figure 3 of the revised PDF).
> * Ablation studies results on $u10$ and $v10$ (see Table 2 of the revised PDF).
> * $u10$ and $v10$ RMSE comparison w.r.t. latitude (see Table 3 of the revised PDF).
>
> We can observe that these results do **not** change our original conclusion and contributions.

---

> > ### Author Response · Authors · 2024-11-19
> > **Author Responses (2/2)**
> >
> > >Weakness 3: The article states that the usage of the Transformer network with Fourier transform is an important component, but both AFNO and FourcastNet use related structures, which is not innovative enough.
> >
> > Thank you for raising the comparison with AFNO and FourcastNet (medium-range forecasting model based on AFNO). We would like to emphasize our technical contributions compared to AFNO as follows:
> >
> > * The objective of AFNO is to design an efficient token mixer for Vision Transformers that can effectively handle high-resolution inputs. Therefore, it designs a lightweight block-diagonal weight matrix to perform frequency domain multiplication. In contrast, CirT performs multi-head attention in the frequency domain to model the interactions among weather patches across various latitudes. The design is distinct from AFNO and innovative.
> > * Moreover, AFNO does not consider geometric inductive bias due to the different motivations from CirT. It employs regular grid patching while CirT introduces a novel framework that is mainly composed of (1) circular patching to standardize patch geometry and (2) a self-attentive frequency mixing strategy to account for spatial periodicity.
> > * We validate our CirT in novel S2S forecasting and empirical results demonstrate it has better generalization ability than FourcastNet.
> >
> > >Questions 3: Most weather models are tuned after training is complete, the article does not perform this work and fine-tuning can be used to further improve model performance.
> >
> > Thanks for providing valuable directions. We agree that fine-tuning after training might improve the model performance and perform ablation studies to evaluate its performance. Specifically, we first train an autoregressive CirT by adapting CirT's output head to forecast next-day weather variables based on the input date (please see the performance of autoregressive CirT in [our response to Reviewer pWHL](https://openreview.net/forum?id=YslOW2SO6S&noteId=OyQSNktohd)). We then freeze the transformer encoder and replace the embedding layers and output head with newly initialized networks to forecast weather variables for Weeks 3-4 and 5-6. The results are as follows:
> >
> > Results of Weeks 3-4 prediction:
> >
> > | RMSE | $z500$ | $z850$ | $t500$ | $t850$ | $t2m$ | $u10$ | $v10$ |
> > | :----|:----|:----|:----|:----|:----|:----|:----|
> > | Fine-tuned decoder  | 540 |346  | 1.885   |   2.327 |  2.715     | 2.013   |   1.619    |
> > |Fine-tuned embedding layer and decoder| 480 | 315 | $\textbf{1.660}$ | $\textbf{1.870}$ | $\textbf{1.983}$ | 1.842 | 1.530 |
> > | Direct training |  $\textbf{477}$| $\textbf{304}$ | 1.687   | 1.903   |    2.007   | $\textbf{1.806}$   | $\textbf{1.511}$  |
> >
> > Results of Weeks 5-6 prediction:
> >
> > | RMSE | $z500$ | $z850$ | $t500$ | $t850$ | $t2m$ | $u10$ | $v10$ |
> > | :----|:----|:----|:----|:----|:----|:----|:----|
> > | Fine-tuned decoder  |588  |  354| 2.190   |  2.702  |  3.145     | 2.043   |   1.650    |
> > |Fine-tuned embedding layer and decoder|485|312|1.679|$\textbf{1.923}$|2.032|1.847|1.535|
> > | Direct training | $\textbf{471}$ | $\textbf{301}$ |$\textbf{1.672}$  | 1.933    |$\textbf{2.026}$       | $\textbf{1.809}$  |   $\textbf{1.512}$ |
> >
> > From the result, we can observe that direct training still performs best in most cases. Meanwhile, we find that fine-tuned embedding layer and decoder improve the performance in several variables such as $t850$. We sincerely thank the reviewer for motivating us to explore fine-tuning CirT which again verified its effectiveness under different training strategies.

---

> > > ### Comment · Reviewer_iiXK · 2024-12-01
> > >
> > > I appreciate the author's explanation of the relevant issues, so I finally updated the score to 6.

---

> > > > ### Author Response · Authors · 2024-12-01
> > > > **Thank you**
> > > >
> > > > Thanks for your reply. We sincerely thank you for lifting the rating.

---

> ### Author Response · Authors · 2024-11-24
> **Looking forward to your feedback**
>
> Dear Reviewer iiXK,
>
> Thanks for your contributions to the reviewing process. As the deadline for the author-reviewer discussion approaches, we kindly request your feedback on whether our responses have satisfactorily addressed your concerns. Should you have any additional suggestions or comments, please do not hesitate to share them with us. We would be more than willing to engage in further discussions and make any necessary improvements.
>
> Thank you once again for dedicating your valuable time to reviewing our work.

---

> ### Author Response · Authors · 2024-11-27
> **A summary of our revision**
>
> Dear reviewer，
>
> Thanks for your valuable comments. As the deadline for revision (closes on November 27th) is fast approaching, we would be grateful if you could allocate some time to review our revision.
>
> We understand that you have a multitude of responsibilities, To facilitate a swift evaluation of our revisions, we have summarized the corresponding changes as follows:
>
> * We add a detailed explanation of how we obtain the results of the Pangu model (Lines 252-260 in the revision).
> * We add the experimental results of wind $u$ and $v$ (Table 1, Table 2, Table 3, and Figure 3 in the revision).
> * We discuss the difference between CirT and FourcastNet in Section 5 (Lines 523-527).
> * We conduct an ablation study to compare the performance of direct training and fine-tuning (Table 5 in the revision).
>
> Please let us know if you have any additional concerns or questions. We kindly request that you re-evaluate our paper based on the provided responses and revision. Thank you for your time and consideration!

---

> ### Author Response · Authors · 2024-11-28
> **Looking forward to your feedback**
>
> Dear Reviewer iiXK,
>
> Thanks for your contributions to the reviewing process. We kindly request your feedback on whether our responses have satisfactorily addressed your concerns. Should you have any additional suggestions or comments, please do not hesitate to share them with us. We would be more than willing to engage in further discussions and make any necessary improvements.
>
> Thank you once again for dedicating your valuable time to reviewing our work.

---

> ### Author Response · Authors · 2024-11-29
> **Looking forward to your reply**
>
> Dear Reviewer iiXK,
>
> Thanks for your comment. Given the substantial work involved, we would appreciate it if you could confirm whether our response has addressed your concerns. If there are any remaining issues, we are willing to engage in further discussion and make additional adjustments.

---

### Official Review · Reviewer_PHHy · 2024-11-01

**Soundness:** 3
**Presentation:** 3
**Contribution:** 3
**Rating:** 6
**Confidence:** 1

**Summary:**

CIRT propose a novel network for accurate S2S climate forecasting. The key idea is to model the cyclic characteristic of the graticule, consisting of two designs: (1) decomposing the weather data by latitude into circular patches; (2) Use fourier transform in self-attention. Extensive experiments on ERA5 demonstrate the effectiveness of the proposed CIRT network.

**Strengths:**

1. The paper is well-written, and the idea and motivation are intuitive. Figure 1 clearly highlights the limitations in previous methods, providing a strong rationale for the proposed approach.

2. CIRT introduces significant innovations to address the lack of geometric inductive biases in prior work. Specifically, it partitions the graticule uniformly by latitude and leverages the Fourier transform to capture global features, which are then processed with self-attention.

3. The experimental results effectively demonstrate the superiority of CIRT, comparing favorably with both numerical and data-driven models.

**Weaknesses:**

GraphCast employs a graph neural network for weather forecasting, sharing a similar initial motivation with CIRT in aiming to reduce inappropriate geometric inductive biases. The authors should consider providing a more in-depth discussion on the distinctions between GraphCast’s model design and system architecture compared to their own, highlighting the unique aspects and advantages of their final approach.

**Questions:**

Add discussion with GraphCast.

---

> ### Author Response · Authors · 2024-11-19
>
> We sincerely appreciate the reviewer for listing the detailed strengths and taking the time to provide feedback. Your support of our work is greatly appreciated. We try our best to eliminate the question via the following response.
>
> >Weakness & Questions: An in-depth discussion on the model design and system architecture distinctions between GraphCast and CirT.
>
> We thank the reviewer for raising the comparison between CirT and GraphCast. We agree that both GraphCast and CirT leverage geometric inductive biases. Nevertheless, they are distinct and we highlight our technical contributions from the following aspects:
>
> * **Motivation**: GraphCast focuses on local state aggregation, probably due to its goal is short-term medium-range forecasting. In contrast, CirT aims to forecast weather in the S2S timescale and focus on capturing the global relations between initial and S2S weather states. Thus, CirT is constructed on a transformer foundation that excels at capturing global information.
> * **Model design**: GraphCast relies on message passing to aggregate local information without explicitly accounting for spatial periodicity. In contrast, CirT employs circular patching to normalize patch geometry and leverages its Fourier representation, consisting of coefficients of periodic basis functions, as inputs to the transformer encoders.
> * **Empirical verification**: Extensive experiments in Table 1 and Figure 3 show that CirT outperforms GraphCast on S2S predictability. Additionally, in Table 3, we can find that CirT outperforms GraphCast in different latitudinal areas, further verifying the effectiveness of our model designs.

---

> ### Author Response · Authors · 2024-11-24
> **Looking forward to your feedback**
>
> Dear Reviewer PHHy,
>
> Thanks for your contributions to the reviewing process. As the deadline for the author-reviewer discussion approaches, we kindly request your feedback on whether our responses have satisfactorily addressed your concerns. Should you have any additional suggestions or comments, please do not hesitate to share them with us. We would be more than willing to engage in further discussions and make any necessary improvements.
>
> Thank you once again for dedicating your valuable time to reviewing our work.

---

> > ### Comment · Reviewer_PHHy · 2024-11-26
> >
> > The authors have addressed my concerns, so I will maintain my original positive rating.

---

> > > ### Author Response · Authors · 2024-11-26
> > > **Thank you**
> > >
> > > Thanks for your reply. We sincerely thank you for your kind feedback.

---

### Official Review · Reviewer_pWHL · 2024-11-02

**Soundness:** 3
**Presentation:** 3
**Contribution:** 3
**Rating:** 6
**Confidence:** 3

**Summary:**

This paper propose an improved approach to the Subseasonal-to-Seasonal (S2S) climate forecasting problem. The method mainly has two novel parts: 1. A better input patching method which utilize the circular nature of the graticule. 2.  Propose to intervene Discrete Fourier Transform throughout the transformer layers.

The propose methods demonstrate improvement over all the existing methods, both numeric methods and data-driven method. And the propose method is especially effective in the long time horizon prediction compared to other autoregressive data-driven methods which suffer from error aggregation. The method also show better performance in the hard-to-predict areas like polar areas.

**Strengths:**

1. The experiments are extensive, solid, and the results are good. The paper compares the proposed method with both numerical method and ml method, and the metrics on latitude-weighted RMSE and Anomaly are strong and solid. The visualization graphs are very useful to understand the impact of the proposed method in different dimensions.

2. The proposed method well utilize the characteristics of the input data modality. The circular patching strategy introduce a effective inductive bias, and the DFT well utilize the latitudinal spatial periodicity.

**Weaknesses:**

1. The comparison to existing method does not include computation complexity/model size. This weakens the comparison to other methods because the reader don't know whether the improvement is brought by the network architecture design or simply due to the increase in model size.

2. It would be a plus if the paper could also include a ablation study of autogressive prediction vs directly predicting all the future values, to ablate the effectiveness of the autoregressive formulation. But I understand this might be consuming.

**Questions:**

I am wondering why the proposed method patch all the different longitudinal areas into the same patch. What prevents the model from taking a 2d input HxWxD and do e.g. axial transformer (intervene between attention along H axis and then W axis) with Fourier Transform.

---

> ### Author Response · Authors · 2024-11-19
>
> We sincerely appreciate the reviewer for listing the detailed strengths and constructive comments. We have made every effort to faithfully address your comments in the following responses.
>
> >Weakness 1: Comparison of model computation complexity and model size.
>
> Thank you for raising the comparison of computation complexity/model size. We compare CirT's Floating point operations (FLOPs) and parameters with two representative models, Graphcast and PanguWeather. The results are as follows:
>
> | Model | Computation Complexity | Model Size |
> | :----|:----|:----|
> | GraphCast | 110 teraFLOPs | 37M |
> | Pangu-Weather | 168 teraFLOPS | 256M |
> | CirT | 2.2 gigaFLOPS | 16M |
>
>
> We can observe that CirT has less computation and smaller model size (due to the lower resolution and embedding size), but achieves better S2S predictivity, verifying our model designs.
>
>
> >Weakness 2: Ablation study of autogressive prediction vs directly predicting all the future values.
>
> Thanks for the instructive suggestion. We adapted CirT's output head to forecast next-day weather variables based on the input date. For inference, it iteratively predicts next-day weather variables up to the S2S timescale. The results are displayed as follows:
>
> Results of Weeks 3-4 prediction:
>
> | RMSE | $z500$ | $z850$ | $t500$ | $t850$ | $t2m$ | $u10$ | $v10$ |
> | :----|:----|:----|:----|:----|:----|:----|:----|
> | Autogressive  |  781| 453 | 3.406|4.014  | 4.584| 2.806 |2.267 |
> | Direct |477| 304| 1.687 |1.903 |2.007  |1.806 | 1.511 |
>
> | ACC | $z500$ | $z850$ | $t500$ | $t850$ | $t2m$ | $u10$ | $v10$ |
> | :----|:----|:----|:----|:----|:----|:----|:----|
> | Autogressive  | 0.962 |0.922  |0.956 |0.957  |0.968 |0.763  |0.610 |
> | Direct |0.984  |0.963  |0.988 |0.988  |0.993 |0.896  | 0.811|
>
> Results of Weeks 5-6 prediction:
>
> | RMSE | $z500$ | $z850$ | $t500$ | $t850$ | $t2m$ | $u10$ | $v10$ |
> | :----|:----|:----|:----|:----|:----|:----|:----|
> | Autogressive  | 813 |455 | 3.636|4.357  |5.047 | 2.855 |2.324 |
> | Direct |471 |301 |1.672  |1.933 |2.026  |1.809 |1.512 |
>
> | ACC | $z500$ | $z850$ | $t500$ | $t850$ | $t2m$ | $u10$ | $v10$ |
> | :----|:----|:----|:----|:----|:----|:----|:----|
> | Autogressive  | 0.960 |0.923  |0.950 |0.949  |0.960 |0.758  | 0.599|
> | Direct | 0.985 | 0.986 | 0.988| 0.989 |0.992 |0.895  |0.812 |
>
> From the results, we can observe that the direct method performs better. The autoregressive CirT still accumulates errors, resulting in inaccurate S2S predictions. Additionally, it underperforms models like PanguWeather and GraphCast which are trained on higher-resolution data (i.e., $0.25^{\circ}$). In contrast, the direct CirT trained on the same dataset (i.e., $1.5^{\circ}$) performs the best, demonstrating its effectiveness.
>
>
> >Question: Why the proposed method patch all the different longitudinal areas into the same patch and do e.g. axial transformer with Fourier Transform?
>
> Thanks for the question and reference. We address the question from the following three aspects:
>
> * Due to the inherent geometric differences between the sphere and the plane, the resulting regular 2D patches will have varying areas and spatial relations. In contrast, circular patch lengths can be determined by their latitudes and the adjacent patches are equidistant, reducing the learning difficulty.
> * Another inductive bias is that the circular patch is spatial periodic. That is, the circular patch satisfies $X_{w} = X_{w + W}$. Therefore, instead of directly inputting $X$ into the transformer, we consider its Fourier transform, which is composed of coefficients from a series of periodic basis functions.
> * Ablation studies in Table 2 show the performance of grid (2D input), grid with Fourier Transform (FT), circular patching, and circular patching with FT. We can observe that circular patching consistently enhances model performance, with optimal results achieved when combined with Fourier transforms, verifying our model designs.

---

> ### Author Response · Authors · 2024-11-24
> **Looking forward to your feedback**
>
> Dear Reviewer pWHL,
>
> Thanks for your contributions to the reviewing process. As the deadline for the author-reviewer discussion approaches, we kindly request your feedback on whether our responses have satisfactorily addressed your concerns. Should you have any additional suggestions or comments, please do not hesitate to share them with us. We would be more than willing to engage in further discussions and make any necessary improvements.
>
> Thank you once again for dedicating your valuable time to reviewing our work.

---

> ### Author Response · Authors · 2024-11-27
> **A summary of our revision**
>
> Dear reviewer，
>
> Thanks for your valuable comments. As the deadline for revision (closes on November 27th) is fast approaching, we would be grateful if you could allocate some time to review our revision.
>
> We understand that you have a multitude of responsibilities, To facilitate a swift evaluation of our revisions, we have summarized the corresponding changes as follows:
>
> * We add the model computation complexity and model size comparison among CirT, GraphCast, and PanguWeather (Table 4 in the revision).
> * We add an ablation study to compare the results of autoregressive prediction and direct prediction (Table 5 in the revision).
> * We add motivation for employing the Fourier Transform (Lines 175-177).
>
> Please let us know if you have any additional concerns or questions. we kindly request your feedback on whether our responses have satisfactorily addressed your concerns. Thank you for your time and consideration!

---

> > ### Comment · Reviewer_pWHL · 2024-11-27
> > **Thank you**
> >
> > Thank you for addressing the comments. The computation complexity studies and ablation on "autoregressive prediction vs direct prediction" look good to me. I will maintain my scores.

---

> > > ### Author Response · Authors · 2024-11-28
> > > **Please kindly consider lifting the ratings**
> > >
> > > Dear reviewer,
> > >
> > > We sincerely appreciate your reply and are glad to hear that our additional results look good.
> > >
> > > Therefore we kindly ask you to consider lifting ratings given the substantial amount of work involved.
> > >
> > > Thanks for your feedback again.
> > >
> > > Best regards,
> > >
> > > Authors.

---

### Author Response · Authors · 2024-11-22
**General Response to All Reviewers**

Dear Reviewers,

We sincerely thank all the reviewers (`pWHL`, `PHHy`, `iiXK`) for their valuable feedback. We are glad that the reviewers appreciated the effectiveness and consistency of our proposed framework (`pWHL`, `PHHy`, `iiXK`), the extensiveness of our experiments (`pWHL`, `PHHy`), the solidness of experimental results (`pWHL`, `PHHy`, `iiXK`), and the overall quality of our paper's writing (`PHHy`, `iiXK`).

We have made every effort to faithfully address your comments in the responses. As suggested by the reviewers, we add

* Additional comparison of model computation complexity and model size ([pWHL](https://openreview.net/forum?id=YslOW2SO6S&noteId=ufTafZz5op)).
* Additional ablation study between autoregressive and direct prediction ([pWHL](https://openreview.net/forum?id=YslOW2SO6S&noteId=ufTafZz5op)).
* Additional results on wind $u$ and $v$ ([iiXK](https://openreview.net/forum?id=YslOW2SO6S&noteId=XVzzP1Of3X)).
* Additional results of fine-tuning CirT ([iiXK](https://openreview.net/forum?id=YslOW2SO6S&noteId=XVzzP1Of3X)).
* Discussion with GraphCast ([PHHy](https://openreview.net/forum?id=YslOW2SO6S&noteId=FB1YN049Ho)) and AFNO ([iiXK](https://openreview.net/forum?id=YslOW2SO6S&noteId=XVzzP1Of3X)).
* Detailed explanation of baseline results ([iiXK](https://openreview.net/forum?id=YslOW2SO6S&noteId=XVzzP1Of3X)) and CirT motivation ([pWHL](https://openreview.net/forum?id=YslOW2SO6S&noteId=ufTafZz5op)).

We have incorporated the suggested modifications in the revised version, which are highlighted in blue. If you are satisfied with the revisions, we kindly request your approval to consider an improved score.

Thanks for all the reviewers' time again.

Best regards,

Authors

---

### Author Response · Authors · 2024-11-25
**A kind reminder: the discussion period will end in two days**

Dear Reviewers,

We sincerely thank you for providing valuable feedback. We kindly remind you that the responses to your concerns have been posted. As the discussion period will end in two days, we would be grateful if you could allocate some time to review our responses.

Thanks for all the reviewers' time again.

Best regards,

Authors

---

### Meta-Review · Area_Chair_f4y3 · 2024-12-17

**Metareview:**

This paper presents a geometry-inspired transformer for numerical weather forecasting. It involves two main improvements: (1) decomposing weather data into patches by geometry, and (2) using Fourier transform. The model performs well in subseasonal-to-seasonal weather forecasting on the ERA5 data.

All three reviewers recommended borderline acceptance, but two of them were not very confident about the recommendation. The AC reads the paper and believes that the paper is of merit to the community, in particular, AI for scientific computing is an important and rising field and ICLR should encourage studies in this direction. The AC recommends acceptance, but leaves the following comments for the authors to address in the final version.
* The comparison may be unfair to other models (FourCastNet, Pangu-Weather, GraphCast) because these models were not optimized for seasonal forecasting and these models have a resolution of 0.25 degrees (which requires more details; but adding more spatial details can be harmful for long-range weather forecasting).
* The authors should clearly mention the relationship between the claimed technical contributions and prior works. The use of geometry (e.g. circular patches) is highly related to the Earth-specific prior of Pangu-Weather, and the Fourier transform is highly related to FourCastNet.
* The work uses 2D neural networks; it is good to discuss whether it is better to use 3D neural networks to integrate various pressure levels (cf. Pangu-Weather, Fengwu, Fuxi, etc.).

**Additional Comments On Reviewer Discussion:**

The reviewers gave diverse scores initially (5/6/8) but converged into the same rating after the rebuttal (6/6/6).

---

### Decision · Program_Chairs · 2025-01-22

Accept (Poster)